# In vivo dissection of a clustered-CTCF domain boundary reveals developmental principles of regulatory insulation

Chiara Anania[1,8], Rafael D. Acemel [1,8], Johanna Jedamzick[1], Adriano Bolondi [2,3], Giulia Cova[4], Norbert Brieske[4], Ralf Kühn [5], Lars Wittler[6], Francisca M. Real [4,7] and Darío G. Lupiáñez [1 ✉]

Vertebrate genomes organize into topologically associating domains, delimited by boundaries that insulate regulatory elements from nontarget genes. However, how boundary function is established is not well understood. Here, we combine genome-wide analyses and transgenic mouse assays to dissect the regulatory logic of clustered-CCCTC-binding factor (CTCF) boundaries in vivo, interrogating their function at multiple levels: chromatin interactions, transcription and phenotypes. Individual CTCF binding site (CBS) deletions revealed that the characteristics of specific sites can outweigh other factors such as CBS number and orientation. Combined deletions demonstrated that CBSs cooperate redundantly and provide boundary robustness. We show that divergent CBS signatures are not strictly required for effective insulation and that chromatin loops formed by non-convergently oriented sites could be mediated by a loop interference mechanism. Further, we observe that insulation strength constitutes a quantitative modulator of gene expression and phenotypes. Our results highlight the modular nature of boundaries and their control over developmental processes.

The development of complex organisms relies on intricate gene expression patterns, resulting from the interaction between distal regulatory elements and genes[1]. High-throughput conformation capture methods (Hi-C)[2,3] revealed that vertebrate genomes organize into topologically associating domains (TADs)[4,5], in which regulatory elements and their target genes are framed[6,7]. TADs are separated by boundary regions that limit the regulatory crosstalk between adjacent domains, and their disruption has been linked to human disease[8–11].

The transcriptional repressor CCCTC-binding factor (CTCF) is found at the majority of boundaries[4] and its depletion leads to a genome-wide disappearance of TADs[12]. At TAD boundaries, the clustering of CBSs with divergent orientation is a conserved molecular signature through vertebrate evolution[13]. The formation of chromatin loops, often associated with TAD boundaries, preferentially occurs between pairs of CBSs displaying convergent motif orientations[14]. This orientation bias is explained by the loop extrusion model, which proposes that the cohesin complex extrudes the chromatin fiber until reaching a CBS in an opposing orientation, but continuing when CTCF is oriented otherwise[15,16].

Although TAD boundaries are fundamental players in the spatial organization of genomes, their influence over developmental gene expression remains controversial. While alterations of TAD boundaries at particular loci can lead to developmental phenotypes[10,17], it only causes moderate transcriptional changes in other genomic regions[18–20]. In addition, the global disruption of TADs in cultured cells results in limited changes in gene expression[12,21]. Furthermore, individual cells can display chromatin conformations

that, in some instances, ignore the TAD boundaries detected in bulk data[22–24]. These contradictory results demonstrate the need for a comprehensive dissection of boundary elements in developmental settings.

Here, we combine genome-wide analyses and mouse genetics to investigate the regulatory logic of clustered-CTCF boundaries in vivo. Using the *Epha4-Pax3* (EP) boundary region as a testbed, we generated 14 mouse homozygous alleles with individual or combined CBS deletions and inversions. Combining capture Hi-C (cHi-C), gene expression and phenotypical analyses, we quantify the functional consequences of boundary perturbations at several levels: ectopic chromatin interactions, gene misexpression and aberrant limb morphologies. Our study reveals fundamental principles of boundary function, delineating a tight interplay between genomic sequence, three-dimensional (3D) chromatin structure and developmental processes.

## Results

**A genetic setup to investigate boundary function in vivo.** We previously demonstrated that a 150-kilobase (kb) region, the EP boundary, is sufficient to segregate the regulatory activities of the *Epha4* and *Pax3* TADs[10] (Extended Data Figs. 1 and 2). The *DelB* background carries a large deletion that removes this boundary region, and the *Epha4* gene, resulting in the ectopic interaction between the *Epha4* limb enhancers and the *Pax3* gene. This causes *Pax3* misexpression and the shortening of fingers (brachydactyly) in mice and in human patients. In contrast, the *DelBs* background carries a similar deletion but not affecting the EP boundary, which maintains

[1]Max-Delbrück Center for Molecular Medicine in the Helmholtz Association (MDC), Berlin Institute for Medical Systems Biology (BIMSB), Epigenetics and Sex Development Group, Berlin, Germany. [2]Department of Genome Regulation, Max Planck Institute for Molecular Genetics, Berlin, Germany. [3]Institute of Chemistry and Biochemistry, Freie Universität Berlin, Berlin, Germany. [4]RG Development & Disease, Max Planck Institute for Molecular Genetics, Berlin, Germany. [5]Max-Delbrück Center for Molecular Medicine in the Helmholtz Association (MDC), Berlin, Germany. [6]Department of Developmental Genetics, Transgenic Unit, Max Planck Institute for Molecular Genetics, Berlin, Germany. [7]Institute for Medical and Human Genetics, Charité – Universitätsmedizin Berlin, Berlin, Germany. [8]These authors contributed equally: Chiara Anania, Rafael D. Acemel. ✉e-mail: Dario.Lupianez@mdc-berlin.de

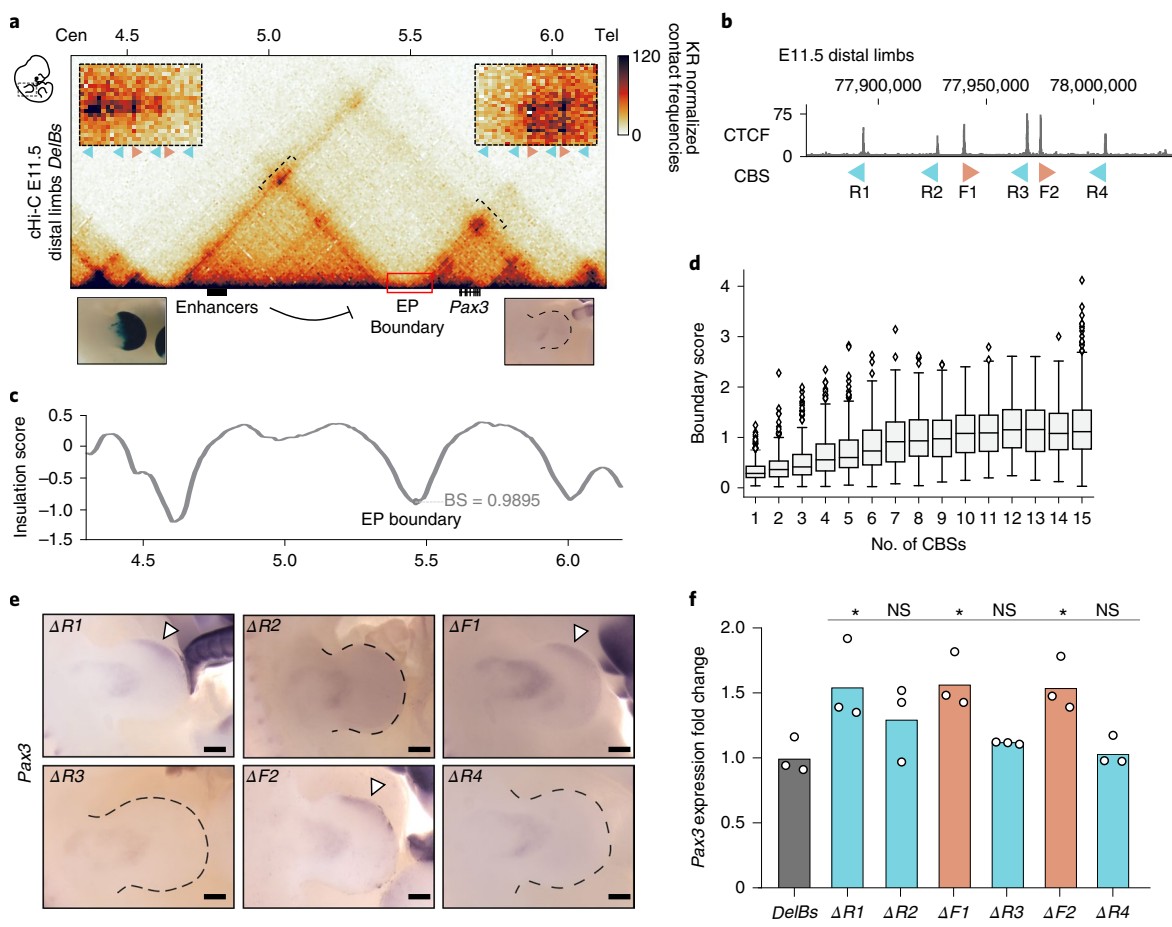

**Fig. 1 | Impact of individual CBS deletions on boundary function. a**, cHi-C maps from E11.5 distal limbs from *DelBs* mutants at 10-kb resolution. Data were mapped on a custom genome containing the *DelBs* deletion (n = 1 with an internal control comparing 6 different experiments; Methods). The red rectangle marks the EP boundary region. Insets represent a magnification (5-kb resolution) of the centromeric (left) and telomeric (right) loops highlighted by brackets on the map. Cen, Centromeric; Tel, Telomeric. Arrowheads represent reverse- (light blue) and forward- (orange) oriented CBSs. Below, Lac-Z staining (left) and WISH (right) of E11.5 mouse forelimbs show activation pattern of *Epha4* enhancers and *Pax3* expression, respectively. **b**, CTCF ChIP–seq track from E11.5 mouse distal limbs. Schematic shows CBS orientation. **c**, Insulation score values. The gray dot represents the local minima of the insulation score at the EP boundary. BS, boundary score. **d**, Relationship between BS and the number of CBSs (data from ref. [26]). The boxes in the boxplots indicate the median and the first and third quartiles (Q1 and Q3). Whiskers extend to the last observation within 1.5 times the interquartile range below and above Q1 and Q3, respectively. The rest of the observations, including maxima and minima, are shown as outliers. N = 8,127 insulation minima found in mESC Hi-C matrices. **e**, WISH shows *Pax3* expression in E11.5 forelimbs from CBS mutants. Note *Pax3* misexpression on the distal anterior region in *ΔR1*, *ΔF1* and *ΔF2* mutants (white arrowheads). Scale bar, 250 μm. **f**, *Pax3* qPCR analysis in E11.5 limb buds from CBS mutants. Bars represent the mean and white dots represent individual replicates. Values were normalized against *DelBs* mutant (ΔΔCt) (two-sided *t*-test *P ≤ 0.05; NS, nonsignificant; P values from left to right: *DelBs* versus *ΔR1*: 0.02; *DelBs* versus *ΔR2*: 0.11; *DelBs* versus *ΔF1*: 0.02; *DelBs* versus *ΔR3*: 0.23; *DelBs* versus *ΔF2*: 0.02; *DelBs* versus *ΔR4*: 0.73). Cen, Centromeric; Tel, Telomeric.

the *Epha4* and *Pax3* TADs and confines the *Epha4* enhancers within their own regulatory domain (Fig. 1a and Extended Data Fig. 1).

To characterize the EP boundary in vivo, we performed CTCF ChIP–seq on developing limbs. This analysis revealed the presence of six clustered CBSs at the EP boundary region (Fig. 1a,b and Extended Data Fig. 2), a profile that is conserved across tissues[25,26]. CTCF motif analyses confirmed the divergent orientation of these sites, a signature of TAD boundaries, with four CBSs in reverse (R) and two in forward orientation (F). Other features associated with boundaries, such as active transcription or housekeeping genes, were not found in the region[27] (Extended Data Fig. 3). cHi-C data from *DelBs* stage E11.5 distal limbs[28] revealed chromatin loops connecting the two forward-oriented CBSs (F1 and F2) with the telomeric boundary of the *Pax3* TAD, and the centromeric boundary of the *Epha4* TAD with the reverse-oriented CBSs R1, R2 and R3 (Fig. 1a,b). However, the close genomic distances between R2 and

F1 and between R3 and F2 preclude the unambiguous assignment of loops to specific sites. RAD21 (cohesin subunit) ChIP–seq experiments in E11.5 distal limbs revealed that R1, F1 and F2, as well as R2 and R3 to a lesser degree, are bound by cohesin (Extended Data Fig. 3), an essential component for the formation of chromatin loops[21,29,30]. These results delineate the EP element as a prototypical boundary region with insulating properties likely encoded and controlled by CBSs.

**CBS characteristics as key determinants of boundary function.** Boundary regions are predominantly composed of CBS clusters[31], suggesting that the number of sites might be relevant for their function. We explored this by calculating boundary scores[32] on available Hi-C maps[26], and categorizing boundaries according to CBS number. We observe that boundary scores increase monotonically with CBS number, reaching a stabilization at ten CBSs (Fig. 1d).

According to this distribution, the EP boundary falls within a range where its function might be sensitive to alterations on CBS number. To test this, we employed a mouse homozygous embryonic stem cell (mESC) line for the *DelBs* background[28], which we edited to generate individual homozygous deletions for each of the six CBSs of the EP boundary region (Supplementary Fig. 1). ChIP–seq experiments revealed that the disruption of the binding motif was sufficient to abolish CTCF recruitment (Supplementary Fig. 2). Subsequently, we employed tetraploid complementation assays to generate mutant embryos and measure the functional consequences of these deletions in vivo[33,34].

Whole-mount in situ hybridization (WISH) on E11.5 mutant embryos revealed that the insulation function of the EP boundary can be sensitive to individual CBS perturbations (Fig. 1e). However, this effect was restricted to CBSs displaying prominent RAD21 binding (*ΔR1*, *ΔF1* and *ΔF2*) (Extended Data Fig. 3). The altered boundary function was evidenced by *Pax3* misexpression on a reduced area of the anterior limb, while the expression domains in other tissues remained unaltered (Supplementary Fig. 3). The disruption of the other CBSs (*ΔR2*, *ΔR3* and *ΔR4*) did not alter *Pax3* expression, demonstrating that the EP boundary can also preserve its function despite a reduction in CBS number.

To quantify *Pax3* misexpression, we performed quantitative PCR (qPCR) in E11.5 forelimbs. Similarly, we observed a modest, but significant, upregulation in *ΔR1*, *ΔF1* and *ΔF2* mutants (Fig. 1f). Importantly, the functionality of individual CBSs is not strictly correlated with CTCF occupancy as the deletion of R3, displaying the highest levels of CTCF binding among the cluster (Fig. 1b and Extended Data Fig. 3), does not result in measurable transcriptional changes (Fig. 1f). Thus, while CBS number influences insulation, the characteristics of individual sites are major determinants of boundary function.

## CBSs cooperate redundantly to provide insulator robustness.
To explore CBS cooperation, we retargeted our *ΔR1* mESC line to generate double knockout mutants with different (*ΔR1 + F2*) or identical CBS orientations (F1 and F2 in *ΔF-all*) (Fig. 2a). WISH revealed an expanded *Pax3* misexpression towards the posterior region of the limb, demonstrating that the EP boundary is compromised in both mutants. Next, we determined the nature of CBS cooperation by qPCR. These experiments revealed that, in both mutants, *Pax3* misexpression exceeded the summed expression levels from the corresponding individual deletions (Fig. 2b). These negative epistatic effects indicate that CBSs are partially redundant, compensating for the absence of each other.

To gain insights on the mechanisms of CBS cooperation, we generated cHi-C maps of the EP locus from E11.5 distal limbs (Fig. 2c and Supplementary Fig. 4). Maps from *ΔR1 + F2* embryos denoted a clear partition between the *EphaA4* and *Pax3* TADs, analogous to *DelBs* control mutants (Fig. 2c). However, subtraction maps revealed decreased intra-TAD interactions for the *Epha4* and *Pax3* TADs, and a concomitant increase in inter-TAD interactions. In addition, we observed the appearance of a loop connecting the outer boundaries of the *Epha4* and *Pax3* TADs (meta-TAD loop; Extended Data Fig. 4)[35]. Accordingly, the boundary score of the EP boundary in *ΔR1 + F2* mutants was decreased, reflecting a weakened structural insulation (Fig. 2d). Virtual Circular Chromosome Conformation Capture (4C) profiles revealed increased chromatin interactions between the *Pax3* promoter and the *Epha4* limb enhancers (Fig. 2e), consistent with the upregulation of *Pax3*. In addition, two of the chromatin loops that connect the EP boundary and the telomeric boundary were abolished, due to the deletion of the F2 anchor and the associated loss of RAD21 (Fig. 2c and Extended Data Figs. 4 and 5). Consequently, the adjacent chromatin loop exhibited a compensatory effect, with increased interactions mediated by the F1 anchor, consistent with higher RAD21 occupancy (Extended Data

Figs. 4 and 5). At the centromeric site, the deletion of R1 causes the relocation of the loop anchor towards an adjacent region containing a reverse-oriented (R2) and the only remaining forward CBS (F1). While the loop extrusion model would predict a stabilization at a reverse CBS[15,16], the short genomic distance between R2 and F1 precludes an unambiguous assignment of the loop anchor. We also observed increased contacts at R3 and R4, suggesting that these sites are functionally redundant.

Then, we examined cHi-C maps from *ΔF-all* mutants, which display a more pronounced *Pax3* misexpression (Fig. 2b). Interaction maps revealed a partial fusion of the *Epha4* and *Pax3* domains (Fig. 2c), accompanied by a notable decrease of the boundary score (Fig. 2d). Virtual 4C profiles confirmed increased interactions between *Pax3* and the *Epha4* enhancers in *ΔF-all* compared with *ΔR1 + F2* mutants, in agreement with the more pronounced *Pax3* upregulation (Fig. 2e). The deletion of all CBSs with forward orientation abolishes the chromatin loops connecting with the telomeric *Pax3* boundary (Fig. 2c and Extended Data Fig. 4). Towards the centromeric side, R1 maintains RAD21 binding and its chromatin loop with the centromeric *Epha4* boundary (Extended Data Figs. 4 and 5). However, other chromatin loops are still discernible and anchored by the R3 and R4 sites, confirming that these sites perform distinct yet partially overlapping functions. These results demonstrate that CBSs can cooperate but also partially compensate for the absence of each other, conferring functional robustness to boundaries.

## Loops formed by nonconvergent CBSs through loop interference.
Chromatin loops are predominantly anchored by CBS pairs with convergent motif orientation[14,36]. Intriguingly, we observed that the combined F1 and F2 deletion (*ΔF-all*) not only disrupts the loops in the expected orientation (telomeric), but also impacts the centromeric one, as observed in the subtraction maps (Fig. 2c). This effect is noticeable at the R2/F1 site, which was associated with a centromeric chromatin loop in the *DelBs* background (Fig. 1a). This demonstrates that the main loop anchor point was not the R2 but the F1 site (Extended Data Fig. 4), suggesting that this CBS can form loops in a nonconvergent orientation. Such mechanism is described by the loop extrusion model, which predicts that loops could create steric impediments that might prevent additional cohesin complexes from sliding through anchor sites[15,16]. This effect would stabilize these additional cohesin complexes, resulting in the establishment of simultaneous and paired nonconvergent and convergent loops (Fig. 3a).

We searched for further biological indications of this mechanism by analyzing ultra-high-resolution Hi-C datasets[26]. First, we identified loop anchors and classified them according to the orientation of their CBS motif and associated loops. Loop anchors were split into convergent-only (only CBSs oriented in the same direction as their anchored loops), nonconvergent (anchor loops in a direction for which they lack a directional CBS) and no-CTCF (no CBS). While most loop anchors belong to the convergent-only category[14,36], 7.6% of them were classified as nonconvergent. Then, we explored whether these nonconvergent loops could be explained by the nonconvergent anchor simultaneously establishing a convergent loop in the opposite direction (Fig. 3a). We calculated the frequency of anchors involved in bidirectional loops for each category and discovered that, while only 5% of convergent-only or no-CTCF anchors participate in bidirectional loops, this percentage increases significantly up to 45% for nonconvergent anchors (Fig. 3b; chi-squared test, $P < 10^{-225}$). To gain further insights into the mechanisms that establish convergent/nonconvergent loop pairs, we calculated the strength of each corresponding paired loop[22]. We observed that the convergent loops linked to a nonconvergent loop are significantly stronger than their nonconvergent counterparts (Fig. 3c,d; Mann–Whitney U-test, $P = 6 \times 10^{-6}$). Next, we explore if convergent

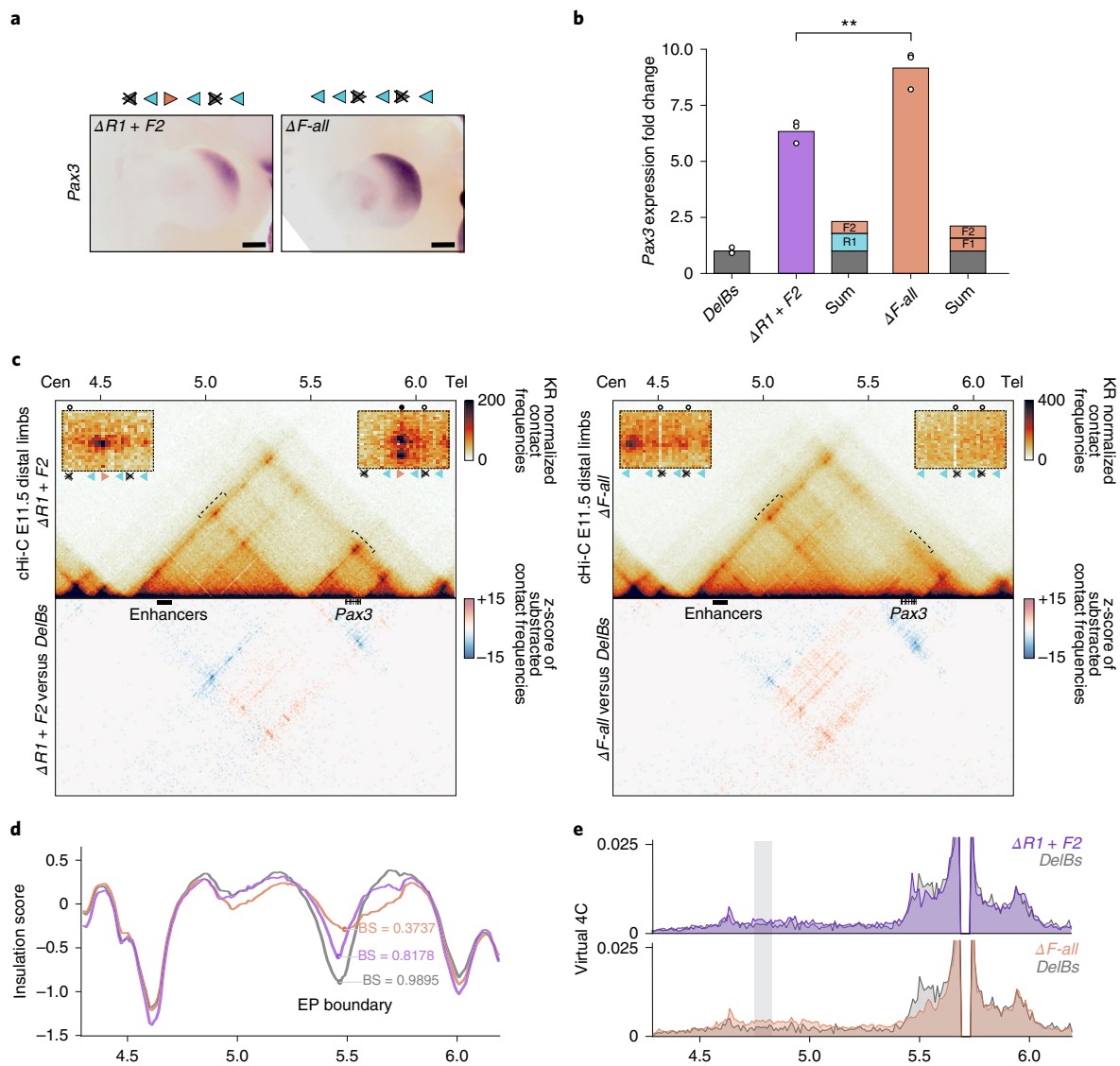

**Fig. 2 | Impact of combined CBS deletions on boundary function. a**, WISH shows *Pax3* expression in E11.5 forelimbs from CBS mutants. Arrowheads represent reverse- (light blue) and forward- (orange) oriented CBSs. Crosses indicate deleted CBSs. Note increased *Pax3* misexpression towards the posterior regions of the limb. Scale bar, 250 μm. **b**, *Pax3* qPCR analysis in E11.5 limb buds from CBS mutants. Bars represent the mean and white dots represent individual replicates. Values were normalized against *DelBs* mutant (ΔΔCt) (**t-test **$P \leq 0.01$; *ΔR1+F2* versus *ΔF-all*: 0.008). **c**, cHi-C maps from E11.5 mutant distal limbs at 10-kb resolution (top). Data were mapped on a custom genome containing the *DelBs* deletion ($n = 1$ with an internal control comparing 6 different experiments; Methods). Insets represent a magnification (5-kb resolution) of the centromeric (left) and telomeric (right) loops highlighted by brackets on the map. Gained or lost chromatin loops are represented by full or empty dots, respectively. Subtraction maps (bottom) showing gain (red) or loss (blue) of interactions in mutants compared with *DelBs*. **d**, Insulation score values. Lines represent indicated mutants. Dots represent the local minima of the insulation score at the EP boundary for each mutant. **e**, Virtual 4C profiles for the genomic region displayed in **c** (viewpoint in *Pax3*). The light-gray rectangle highlights the *Epha4* enhancer region. Note increased interactions between the *Pax3* promoter and the *Epha4* enhancer in *ΔR1+F2* and *ΔF-all* (purple and orange) compared with *DelBs* mutants (gray).

loops paired to nonconvergent loops are particularly strong in comparison with other types of convergent loops. This analysis revealed that the strength of these convergent loops is similar to other unpaired convergent loops across the genome (Extended Data Fig. 6; single-sided convergent category). However, paired convergent/nonconvergent loops appear to be mechanistically different from unpaired loops, as they are more often associated with TAD corners (Extended Data Fig. 6c; chi-squared test, $P < 3.5 \times 10^{-6}$) and therefore connect anchor points that are located farther away in the linear genome (Extended Data Fig. 6d; Mann–Whitney $U$-test, $P < 4.8 \times 10^{-8}$). A comparison against pairs of convergent/convergent loops, which are similarly associated with TAD corners

(Extended Data Fig. 6b; category double-sided convergent), revealed that the convergent loops in convergent/nonconvergent pairs are on average stronger (Mann–Whitney $U$-test, $P = 7 \times 10^{-5}$). This type of convergent/nonconvergent loops can be observed at relevant developmental loci, such as the *Osr1*, *Ebf1* and *Has2* loci (Extended Data Fig. 7). Overall, our analyses suggest that a considerable number of nonconvergent loops could be mechanistically explained by the presence of a stronger and convergent chromatin loop in the opposite orientation and anchored by the same CBS.

To validate these findings in vivo, we sequentially retargeted our *ΔR1* mESCs to create a mutant that only retains the forward F1 and F2 sites, which have strong functionality (Fig. 2a,b). During the

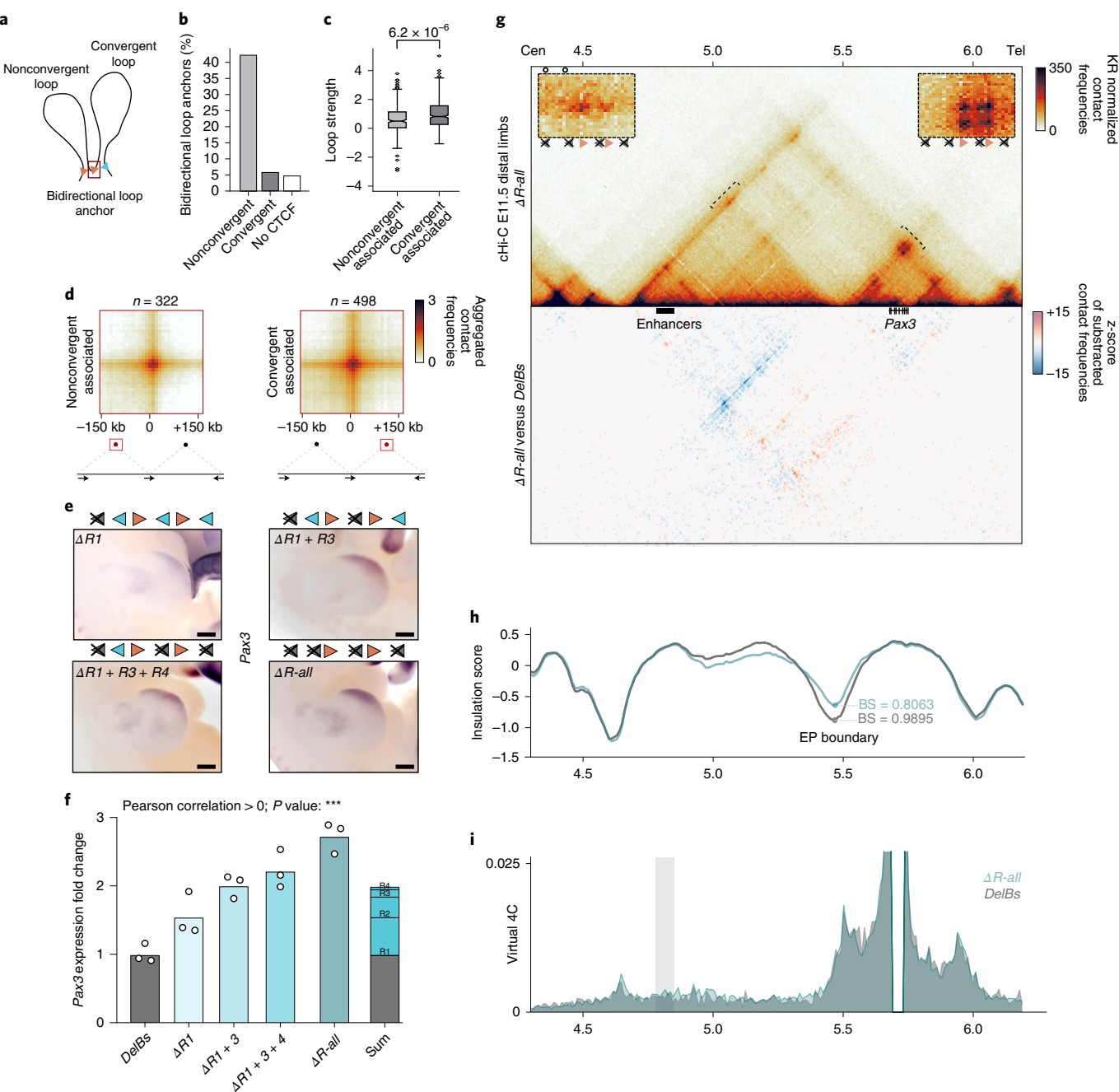

**Fig. 3 | Formation of chromatin loops by nonconvergently oriented CBSs. a**, Schematic of a convergent loop that indirectly generates a nonconvergent loop in the opposite direction. **b**, Percentage of loop anchors establishing bidirectional loops ($n = 12,635$ loops from mESCs from ref. [26]). Anchor categories: convergent-only (only CBSs oriented in the same direction as their anchored loops, $n = 7,769$), nonconvergent (anchor loops in a direction for which they lack a directional CBS, $n = 960$) and no-CTCF (no CBS, $n = 3,906$). **c**, Loop strengths in pairs of convergent/nonconvergent loops classified into Non-conv.-associated (nonconvergent loop sharing the nonconvergent anchor with a convergent loop in the opposite direction, $n = 322$) and Conv.-associated (convergent loop sharing one anchor with a nonconvergent loop in the opposite direction, $n = 496$). Boxplots defined as in Fig. 1c. Two-sided Benjamini–Hochberg-corrected Mann–Whitney $U$-test $P = 6.2 \times 10^{-6}$. **d**, Aggregated loop signal for categories in **c**. Arrows represent CBS orientation. **e**, *Pax3* WISH in E11.5 forelimbs from CBS mutants. Arrowheads represent reverse- (blue) and forward- (orange) oriented CBSs. Crosses indicate deleted CBSs. Note the positive correlation between expanded *Pax3* misexpression and increased number of deleted CBSs. Scale bar, 250 μm. **f**, *Pax3* qPCR analysis in E11.5 limbs from CBS mutants. Bars represent mean and dots individual replicates. Values were normalized against *DelBs* mutant ($\Delta\Delta Ct$). Note the positive correlation of *Pax3* misexpression with the increase in deleted CBSs (Pearson correlation significantly > 0; ***$P \le 0,001$). **g**, cHi-C maps from E11.5 mutant distal limbs at 10-kb resolution (top). Data were mapped on a custom genome containing the *DelBs* deletion ($n = 1$ with an internal control comparing 6 different experiments; Methods). Insets represent a magnification (5-kb resolution) of the centromeric (left) and telomeric (right) loops highlighted by brackets on the map. Gained or lost chromatin loops are represented by full or empty dots. Subtraction maps (bottom) showing gain (red) or loss (blue) of interactions in mutants compared with *DelBs*. **h**, Insulation score values. Dots represent the local minima of the insulation score at the EP boundary for each mutant. **i**, Virtual 4C profiles for the region in **g** (viewpoint in *Pax3*). The gray rectangle highlights *Epha4* enhancers. Note increased interactions between the *Pax3* promoter and the *Epha4* enhancers in *ΔR-all* compared with *DelBs*.

process, we obtained intermediate mutants with double ($\Delta R1 + R3$) and triple CBS deletion ($\Delta R1 + R3 + R4$), as well as the intended quadruple knockout lacking all reverse CBSs ($\Delta R\text{-}all$). WISH revealed an expanded *Pax3* expression pattern towards the posterior limb region, an effect that increases with the number of deleted CBSs (Fig. 3e). Expression analyses by qPCR confirmed a significant increasing trend in *Pax3* misexpression levels across mutants (Fig. 3f; Pearson correlation > 0, $P \leq 2 \times 10^{-7}$). These results demonstrate again that R2, R3 and R4 are functionally redundant sites, despite the absence of measurable effects upon individual deletions (Fig. 1b). However, we noted that *Pax3* levels were only moderately increased (threefold) compared with the expression in mutants retaining only-reverse CBSs (ninefold, $\Delta F\text{-}all$). Importantly, $\Delta R\text{-}all$ mutants retain two intact CBSs in the forward orientation, while up to four CBSs are still present in $\Delta F\text{-}all$ mutants, suggesting that these two forward CBSs (F1 and F2) grant most of the insulator activity of the EP boundary. These experiments indicate that the functional characteristics of specific CBSs can outweigh other predictive parameters of boundary function such as the total number of sites.

As expected, cHi-C maps from $\Delta R\text{-}all$ mutant limbs revealed a clear partition between the *Epha4* and *Pax3* TADs (Fig. 3g), consistent with the reduced *Pax3* misexpression. Boundary scores at the EP boundary were also only moderately reduced (Fig. 3h), in comparison with the broader effects of the $\Delta F\text{-}all$ mutant (Fig. 2d). Accordingly, intra-TAD interactions modestly decreased while inter-TAD interactions increased, as also observed in virtual 4C profiles (Fig. 3i). Despite the multiple deletions, the telomeric chromatin loops remained unaffected and anchored by the F1 and F2 sites, both occupied by RAD21 (Fig. 3g and Extended Data Figs. 4 and 5). However, we noticed the persistence of centromeric chromatin loops anchored by the F1 and F2 sites, despite their nonconvergent forward orientation. A higher contact intensity is observed at F1, which would be the first CBS encountered by cohesin complexes sliding from the centromeric side (Extended Data Figs. 4 and 5).

Finally, we investigated if the formation of nonconvergent loops might be associated with the accumulation of cohesin complexes over a limited number of CBSs. We generated a mutant that only retains the R3 CBS (*R3-only*), which is prominently bound by CTCF (Fig. 1b). We hypothesized that, in the absence of others, this CBS may accumulate the cohesin and form a nonconvergent loop. However, although R3 was the only site able to stall cohesin in this background (Extended Data Fig. 4), cHi-C maps revealed a single convergent loop towards the centromeric side (Extended Data Fig. 8). This loop displays a weak insulator function, denoted by a decreased boundary score, an *Epha4* and *Pax3* TAD fusion and prominent *Pax3* misexpression. Therefore, our results in transgenic mice support our findings at the genome-wide level (Fig. 3a–c), demonstrating that specific CBSs can create chromatin loops independently of their motif orientation, seemingly through loop interference.

**Divergent CBSs are not required for robust insulation.** Previous studies identified divergent CBS clusters as a signature of TAD boundaries, suggesting a role on insulation[13,31]. While our analysis on mutants with reverse-only CBS orientation ($\Delta F\text{-}all$) showed a severe impairment of boundary function (Fig. 2c), this was not the case for $\Delta R\text{-}all$ mutants, which retain CBSs only in the forward orientation (Fig. 3f). Indeed, the levels of *Pax3* misexpression evidenced that insulation is more preserved in $\Delta R\text{-}all$ than in $\Delta R1 + F2$ mutants, which still conserve a divergent CBS signature (Fig. 2c).

This prompted us to explore the relation between CBS composition at boundaries and insulation strength. We examined available Hi-C datasets, classifying boundary regions according to different parameters of CBS composition (that is, number and orientation) and calculating boundary scores (Fig. 4a). Our analysis revealed that, for the same CBS number, boundaries with divergent

signatures generally display more insulation than their nondivergent counterparts. However, up to 6% of nondivergent boundaries display scores above 1.0, a value associated with robust functional insulation (Fig. 1c). Manual inspection at specific loci showed that nondivergent boundaries with strong boundary scores present clear TAD partition and no evidence of coregulation for genes located at either side (Extended Data Fig. 9). These results suggest that a divergent signature is not strictly required to form strong functional boundaries.

**Boundary orientation has a limited impact on insulation.** Next, we explored if the genomic contexts might explain the prominent insulation differences between only-reverse ($\Delta F\text{-}all$) or only-forward ($\Delta R\text{-}all$) mutants. To evaluate this, we generated a mutant with a homozygous inversion of the boundary region, on the $\Delta F\text{-}all$ background ($\Delta F\text{-}all\text{-}Inv$) (Fig. 4b and Supplementary Fig. 5).

WISH and qPCR experiments showed that *Pax3* expression is almost indistinguishable from the $\Delta F\text{-}all$ mutants, both spatially and at the quantitative level (Fig. 4b,c). Moreover, cHi-C maps from $\Delta F\text{-}all\text{-}Inv$ mutants revealed a similar fusion of the *Epha4* and *Pax3* TADs (Fig. 4d). However, subtraction maps showed a redirection of chromatin loops, which now interact mainly with the telomeric *Pax3* boundary instead of the centromeric *Epha4* boundary. These ectopic loops are mainly anchored by the R1 site, which preserves its marked functionality. Despite these local differences, boundary scores and virtual 4C profiles remained comparable between $\Delta F\text{-}all\text{-}Inv$ and $\Delta F\text{-}all$ mutants (Fig. 4e,f). These results suggest that the orientation of entire boundary regions, as well as the differences in the surrounding genomic context, play a minor role in insulator function.

**Genomic distances can influence gene expression levels.** To determine to what extent CTCF binding contributes to the EP boundary function, we generated a sextuple knockout with all CBSs deleted ($\Delta ALL$). WISH revealed a further expansion of *Pax3* misexpression, covering the distal limb entirely. This expanded expression mirrors that of *DelB* mutants, in which the entire boundary region is deleted (Fig. 5a). Expression analyses revealed that *Pax3* misexpression in $\Delta ALL$ mutants exceeds the combined sum of expression from $\Delta R\text{-}all$ and $\Delta F\text{-}all$ mutants (Fig. 5b), again indicating the cooperative and redundant CBS action. Intriguingly, *Pax3* misexpression in the *R3-only* background was comparable to $\Delta ALL$, suggesting that a functionally weak CBS is not sufficient to hinder enhancer–promoter communication (Extended Data Fig. 8). Nevertheless, $\Delta ALL$ mutants only reach 65% of the *Pax3* misexpression observed in *DelB* mutants (Fig. 5b), which may be attributed to the 150-kb inter-CBS region that differentiates both mutants.

To investigate the reduced *Pax3* misexpression in $\Delta ALL$, compared with *DelB* mutants, we performed cHi-C experiments (Fig. 5c). These experiments revealed a prominent *Epha4* and *Pax3* TAD fusion, with increased intensity of the meta-TAD loop (Extended Data Fig. 4). This results from the severe disruption of the EP boundary, denoted by a reduced boundary score (Fig. 5d) and the complete absence of RAD21 binding or anchored loops (Extended Data Figs. 4 and 5). In fact, the interaction profile at the EP boundary is not different from other internal locations of the *Epha4* TAD (Fig. 5c). Of note, higher insulation is observed in *R3-only* compared with $\Delta ALL$, despite the comparable *Pax3* misexpression between both genetic backgrounds (Extended Data Fig. 8). However, virtual 4C profiles from $\Delta ALL$ and *R3-only* mutants confirmed a similar interaction between *Epha4* enhancers and *Pax3* (Fig. 5e and Extended Data Fig. 8). These enhancer–gene interactions were reduced in comparison with *DelB*, in which *Pax3* misexpression is more prominent (Fig. 5e and Extended Data Fig. 8). ChIP–seq datasets for epigenetic marks did not reveal additional regions with regulatory potential within the 150-kb region (Extended Data Fig. 3), indicating that the

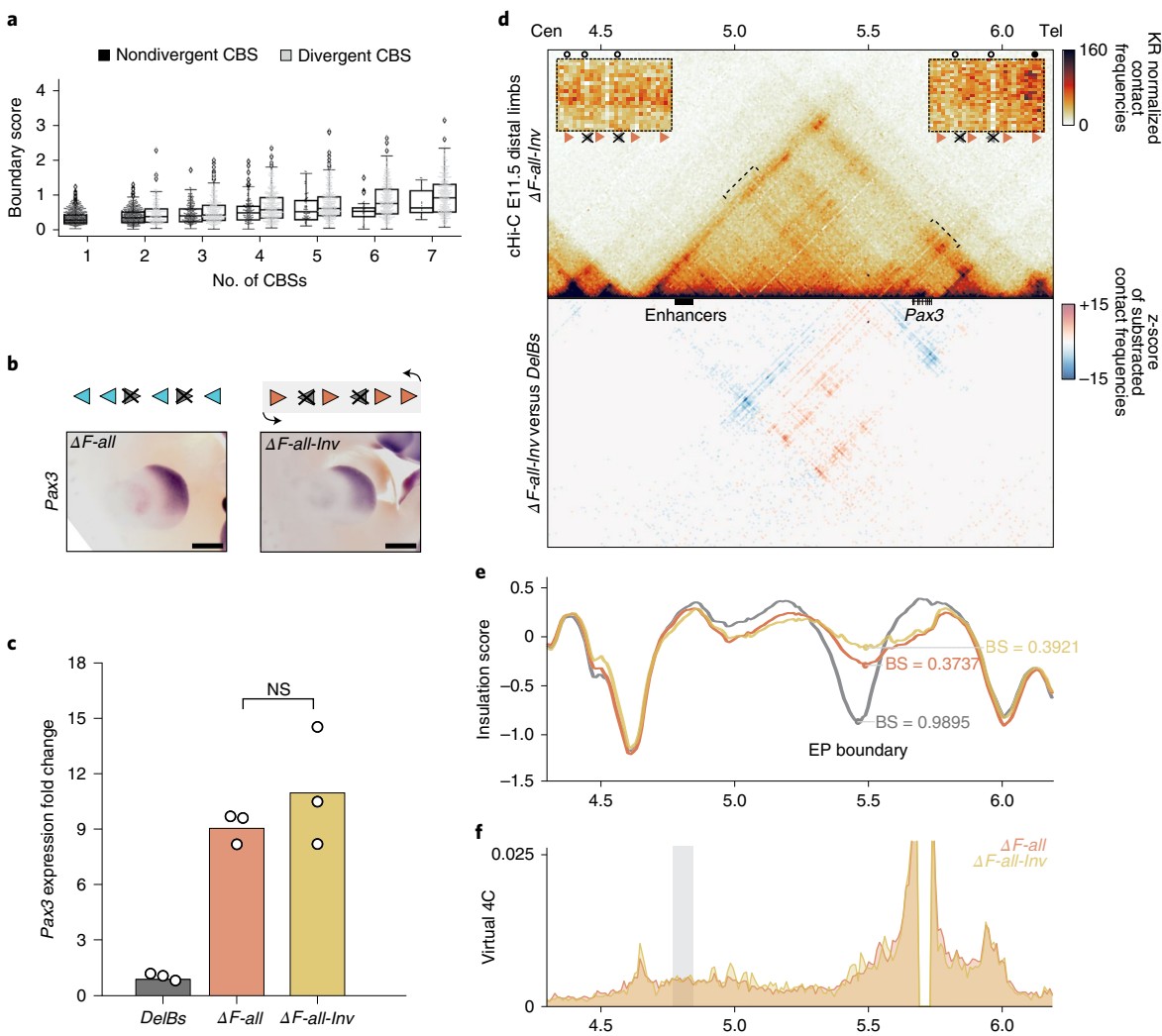

**Fig. 4 | Nondivergent boundary signatures and effects of surrounding genomic context. a**, Relation between BSs and the number of CBSs for divergent and nondivergent boundaries in mESC Hi-C data[26]. Boxplots defined as in Fig. 1c. **b**, WISH shows *Pax3* expression in E11.5 forelimbs from CBS mutants. Arrowheads represent reverse- (light blue) and forward- (orange) oriented CBSs. Crosses indicate deleted CBSs. Light-gray rectangle marks inverted region. Note similar *Pax3* misexpression pattern between *ΔF-all-Inv* and *ΔF-all* mutants. Scale bar, 500 μm. **c**, *Pax3* qPCR analysis in E11.5 limb buds from CBS mutants. Bars represent the mean and white dots represent individual replicates. Values were normalized against *DelBs* mutant (ΔΔCt) (two-sided *t*-test *P* value). **d**, cHi-C maps from E11.5 mutant distal limbs at 10-kb resolution (top). Data mapped on custom genome containing the *DelBs* deletion and the inverted EP boundary (n = 1 with an internal control comparing 6 different experiments; Methods). Insets represent a magnification (5-kb resolution) of the centromeric (left) and telomeric (right) loops highlighted by brackets on the map. Gained or lost chromatin loops are represented by full or empty dots, respectively. Subtraction maps (bottom) showing gain (red) or loss (blue) of interactions in mutants compared with *DelBs*. **e**, Insulation score values. Lines represent mutants. Dots represent the local minima of the insulation score at the EP boundary for each mutant. **f**, Virtual 4C profiles for the genomic region displayed in **d** (viewpoint in *Pax3*). Light-gray rectangle highlights *Epha4* enhancer region. Note similar interaction profile between *ΔF-all-Inv* (yellow) and *ΔF-all* mutants (orange).

enhanced *Pax3* misexpression in *DelB* mutants is unlikely caused by the deletion of regulatory elements. Taken together, these results suggest that enhancer–promoter distances might influence gene expression levels.

**Boundary insulation modulates gene expression and phenotypes.**
*PAX3* misexpression during limb development can cause shortening of thumb and index finger (brachydactyly), in human patients and mouse models[10]. Therefore, our mutant collection provides an opportunity to study how boundary insulation strengths translate into developmental phenotypes.

We obtained mutant E17.5 fetuses and performed skeletal stainings, measuring relative digit length as a proxy for the phenotype

(Fig. 6a,b). First, we analyzed *ΔR1* mutants, which displayed moderate *Pax3* misexpression in the anterior distal limb (Fig. 1f). Finger length ratios revealed that *ΔR1* limbs develop normally, demonstrating that the detrimental effects of *Pax3* misexpression can be partially buffered.

In contrast, *ΔR1 + F2* mutants displayed a moderate reduction of index digit length (Fig. 6a,b), consistent with their increased *Pax3* misexpression (Fig. 2b). This demonstrates that weakened boundaries can be permissive to functional interactions between TADs, resulting in altered transcriptional patterns and phenotypes. Importantly, the phenotypes of *ΔR1 + F2* mutants occur despite an observable partition between *Epha4* and *Pax3* TADs and across a boundary region displaying high boundary scores (Fig. 2c,d;

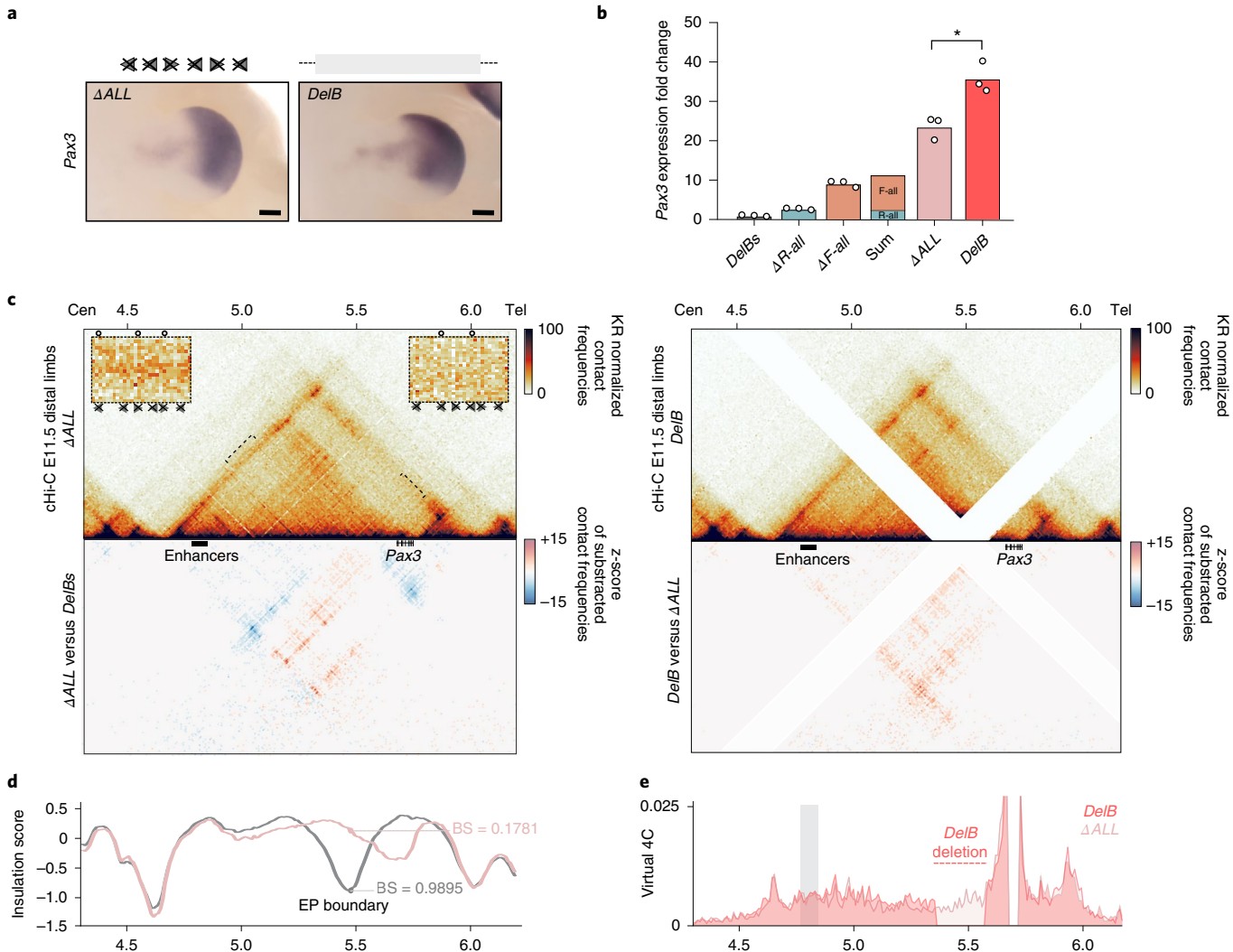

**Fig. 5 | Contribution of CTCF binding to the insulation function of the EP boundary. a**, WISH shows *Pax3* expression in E11.5 forelimbs from CBS mutants. Arrowheads represent reverse- (light blue) and forward- (orange) oriented CBSs. Crosses indicate deleted CBSs and the gray rectangle represents the deleted region. Note the similarities in expression pattern between mutants. Scale bar, 250 μm. **b**, *Pax3* qPCR analysis in E11.5 limb buds from CBS mutants. Bars represent the mean and white dots represent individual replicates. Values were normalized against *DelBs* mutants (ΔΔCt) (*two-sided *t*-test $P \leq 0.05$, *ΔALL* versus *DelB*: 0.03). **c**, cHi-C maps from E11.5 mutant distal limbs at 10-kb resolution (top). Data mapped on custom genome containing the *DelBs* deletion ($n = 1$ with an internal control comparing 6 different experiments; Methods). Insets represent a magnification (5-kb resolution) of the centromeric (left) and telomeric (right) loops highlighted by brackets on the map. Gained or lost chromatin loops represented by full or empty dots, respectively. Subtraction maps (bottom) showing gain (red) or loss (blue) of interactions in mutants compared with *DelBs* (left) and *DelB* (right). **d**, Insulation score values. Lines represent mutants. Dots represent the local minima of the insulation score at the EP boundary for each mutant. **e**, Virtual 4C profiles for the genomic region displayed in **c** (viewpoint in *Pax3*). Light-gray rectangle highlights *Epha4* enhancer region.

boundary score = 0.8). Analyses on ultra-high-resolution Hi-C datasets[26] revealed that many boundary scores fall within the ranges described in our mutant collection (Extended Data Fig. 10). Of note, 40% of boundaries display scores lower than 0.8. According to our observations, those boundaries could be permeable for functional interactions across domains.

Finally, we analyzed the *ΔF-all* mutants, in which the *Epha4* and *Pax3* TADs appear largely fused (Fig. 2c). This disruption of TAD organization led to a prominent reduction of digit length (Fig. 6a,b), consistent with the higher *Pax3* misexpression (Fig. 2b). Overall, these results illustrate how boundary insulation strength can modulate gene expression and developmental phenotypes, by allowing permissive functional interactions between TADs.

## Discussion

Our study reveals principles of boundary function in vivo, demonstrating that CBSs act redundantly and cooperate to establish precise regulatory insulation. On the one hand, the EP boundary function was increasingly compromised with the number of CBS mutations and remained almost unaffected upon individual deletions, as also reported for the *Shh* locus[19,20]. This was similarly reported at the *Rhbdf1/alpha-globin* locus, in which double CBS deletions increase *Rhbdf1* expression by tenfold compared with individual mutations[37]. On the other hand, combined mutants carrying F1 or F2 CBS deletions resulted in enhanced *Pax3* misexpression, thus escaping the general additive trend observed for consecutive mutations of reverse-oriented CBSs (Fig. 6c). Therefore, boundary function appears to be largely determined by the characteristics of

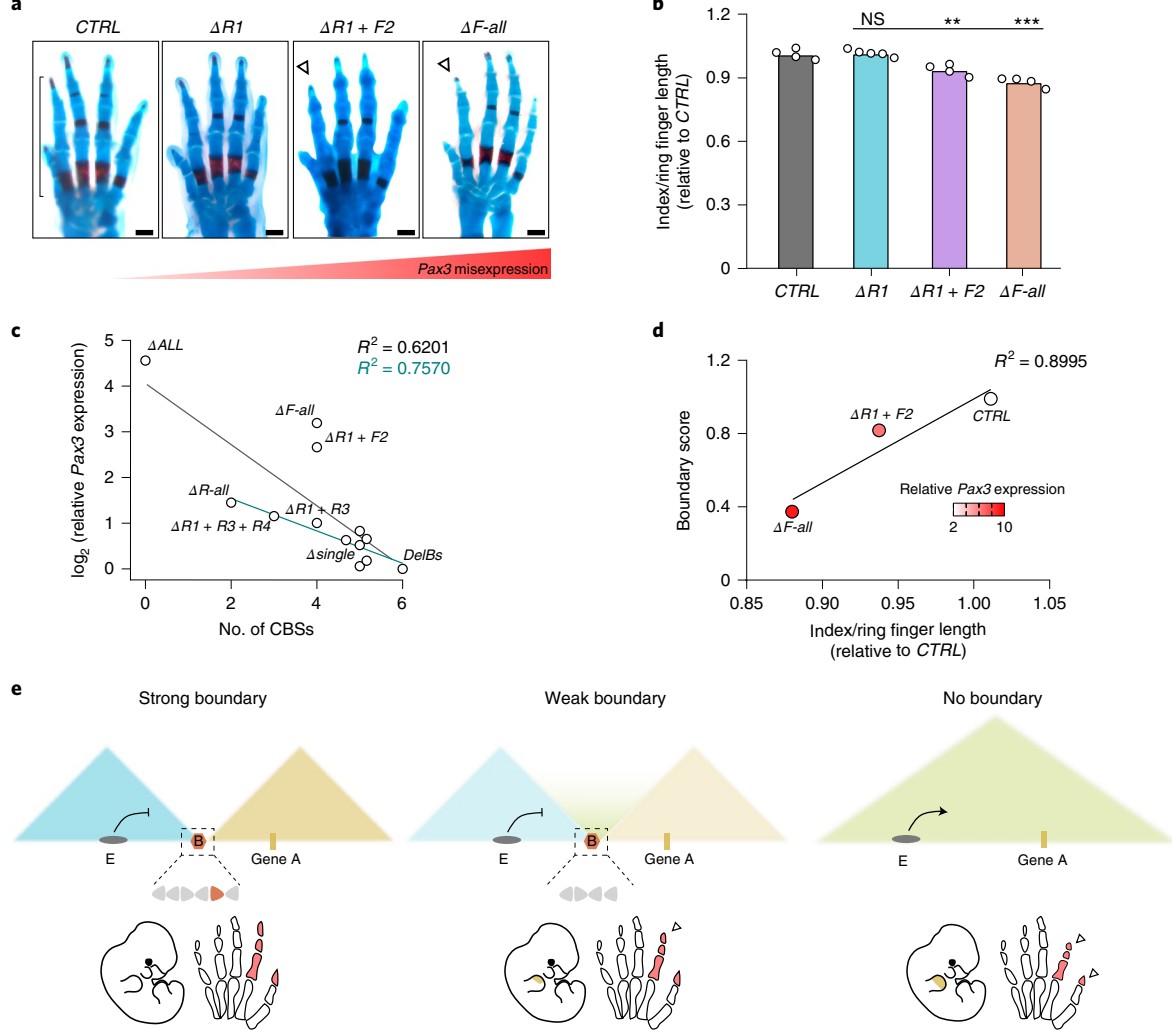

**Fig. 6 | Boundary strength modulates developmental phenotypes. a**, Skeletal staining of forelimbs from E17.5 mutant and control fetuses. White arrowheads indicate reduced index finger lengths. Black bracket shows the region of the finger measured for the quantification. Finger length correlates negatively with increased *Pax3* misexpression. Scale bar, 500 μm. **b**, Index lengths relative to ring finger lengths in E17.5 mouse forelimbs. Bars represent the mean and white dots represent individual replicates. Values were normalized on control (CTRL) animals (two-sided *t*-test **$P \le 0.01$; two-sided *t*-test ***$P \le 0.001$; *ΔR1 + F2* versus CTRL: 0.007; *ΔF-all* versus CTRL: 0.0002). **c**, Correlation between the number of remaining CBSs at the EP boundary and the levels of *Pax3* expression in the different mutants described in this study. Pearson regression lines are shown together with $R^2$ values, both for the whole collection of mutants (black) and discarding combined CBS deletions involving CBSs with forward orientation (turquoise). **d**, Correlation and $R^2$ between BSs and the brachydactyly phenotype penetrance measured as the index to ring finger length ratio for controls, *ΔR1 + F2* and *ΔF-all* mutants. The color of the dots represents the level of *Pax3* limb misexpression as measured by qPCR. **e**, Model for boundary insulation as a quantitative modulator of gene expression and developmental phenotypes. Left, a strong boundary (B) efficiently insulates gene A from the enhancers located in the adjacent TAD (E). The boundary shows a cluster of CBSs with different orientations represented with arrowheads. The colored arrow represents a CBS with prominent contribution to boundary function. Middle, the absence of specific CBSs results in a weakened boundary, moderate gene misexpression (limb, indicated in yellow) and mild phenotypes (reduced digits, indicated in red and pointed out by white arrowhead). Right, the absence of the boundary causes a fusion of TADs, strong gene misexpression and strong phenotypes.

specific CBSs, rather than by their total number or orientation. The differences in CBS function often correlate with CTCF occupancy although with prominent exceptions, such as R3, suggesting that additional factors modulate CBS function. An in vitro insulator reporter approach revealed that flanking genomic regions can also contribute to CBS function, potentially serving as binding platforms for such modulators[38].

Interestingly, the latter study also demonstrates that CBSs can act synergistically[38]. In contrast, we observe that CBSs can also compensate for the absence of each other. These results may suggest that synergistic effects are negligible when the number of clustered CBSs increases. This functional redundancy also seems to converge with additional buffering mechanisms that confer transcriptional and phenotypic robustness against genetic perturbations. This is exemplified by the moderate *Pax3* misexpression of a partial boundary disruption, which is not sufficient to cause abnormal phenotypes (Fig. 6a). Therefore, developmental phenotypes are controlled by complementary 'fail-safe' mechanisms operating at multiple levels: the redundancy of noncoding elements, such as enhancers and insulators, combined with downstream mechanisms that buffer fluctuations in gene expression. Understanding how these mechanisms are interconnected will help to predict which TAD boundary perturbations might cause developmental phenotypes (as described here) or moderate transcriptional effects[19,20].

Our study also highlights that divergent CBS signatures are not strictly required for robust boundary function. This has been also reported at the mouse *HoxD* boundary, where the deletion of all forward-oriented CBSs did not cause a TAD fusion[39]. Nondivergent boundaries can also display boundary scores above those reported to be functionally robust for the EP boundary. Many of those nondivergent boundaries are formed by nonconvergent loops paired to convergent ones. Such configuration can be explained by loop interference, where the persistent anchoring of cohesin might stall additional complexes. A similar phenomenon, termed loop collision, has been observed in cells depleted of the cohesin-releaser factor *WAPL* and, to a lesser degree, in wild-type cells[40]. Our results extend those findings, constituting an in vivo experimental validation for a scenario predicted by the loop extrusion model[15,16].

Single-cell Hi-C[22,41] and super-resolution microscopy[24,42] demonstrated that chromatin interactions in individual cells appear to be stochastic and can even ignore the presence of well-defined boundaries in bulk data. In light of these studies, our results reinforce the premise that boundary insulation should be considered as a quantitative property, as enhancer–promoter crosstalk and gene activation correlate with the strength of structural insulation at boundaries (Fig. 6d). Nevertheless, our comparison between *ΔALL* and *R3-only* mutants suggests that, below certain thresholds, subtle differences in insulation might be insufficient to alter enhancer–promoter communication (Extended Data Fig. 8). These effects seem to depend on the nature of such enhancer–promoter interactions, as demonstrated through CBS insertions at the *Sox2* locus[43]. Besides boundary insulation, we observe that enhancer–promoter communication may be influenced by genomic distances. Indeed, analyses in mESCs demonstrated that the transcriptional output depends on the genomic distance between an enhancer and its promoter[44]. Moreover, reduced distances between the ZRS enhancer and the *Shh* gene, in inverted alleles, can overcome boundary insulation and cause gene activation[45]. Therefore, insulation strength emerges as a key feature of boundary function, which can effectively modulate gene activation and phenotypes (Fig. 6e). In summary, we show that chromatin boundaries are modular and constitute multicomponent genomic regions subjected to several regulatory principles. These principles help to bridge the gap between 3D genome structures and developmental processes.

## Online content

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

## Methods

Research carried out in this study complies with all relevant ethical regulations for animal experimentations. Mice were handled according to institutional guidelines, under the experimentation license (G0111/17) approved by Landesamt für Gesundheit und Soziales (Berlin, Germany). Data collection and analysis were not performed blind to the conditions of the experiments. Samples were excluded only according to the genotype assessed in control experiments.

**Generation of CBS mutant mESCs and transgenic mice.** Mutant mESCs were obtained following an already described method[34]. All CBS deletions were generated using only one single guide RNA (sgRNA) designed in proximity to the binding motif using the website Benchling (https://www.benchling.com/), with the only exceptions of ΔR2 and ΔF-all-Inv which were generated by using a pair of sgRNAs. The guide sequences are listed in Supplementary Table 1. The targeted DelBs mESCs were previously derived from a homozygous DelBs mouse line at the Max Planck Institute for Molecular Genetics, Berlin[28]. Each clone was genotyped by PCR (MangoTaq, Bioline, Cat. No. 25033) (primers listed in Supplementary Table 1). To obtain combined CBS deletions, individual CBS mutants were retargeted following the same procedure.

The engineered mESCs were successively used to generate embryos by tetraploid aggregation methods, as previously described[33,34]. CD-1 and NMRI, females and males of various ages, were used as donors and fosters for embryo retransferring by tetraploid aggregation. The specimens isolated to perform experimental analysis were Bl6/129Sv5 male, E11.5 and E17.5 in age. All mice were housed in standard cages at the Animal Facilities of the Max Planck Institute for Molecular Genetics and the Max-Delbrück-Center for Molecular Medicine in Berlin in a pathogen-free environment. Mice were exposed to Type II Blue Line (base area of 536 cm²) or Type II Green Line (base area 501 cm²) and kept at 22 ± 2 °C with 55 ± 10% humidity and a 12-h light regimen.

**WISH and skeletal preparations.** WISH was performed in wild-type and mutant E11.5 embryos ($n \geq 3$) according to standard procedures. Pax3 probes were generated by PCR amplification using mouse limb bud complementary DNA. For skeletal preparations, wild-type and mutant E17.5 embryos ($n \geq 4$) were stained with Alcian blue/Alizarin red according to standard protocols.

**qPCR.** E11.5 forelimb buds from mutant embryos (male Bl6/129Sv5) were dissected in 1× PBS, collected and snap-frozen in liquid nitrogen. Tissue was dissolved in RLT with the help of syringes and RNA extracted following the guidelines of the RNeasy Mini Kit (Qiagen). Reverse transcription was performed using Applied Biosystems High-Capacity cDNA Reverse Transcription Kit (Cat. No. 4368814) following the manufacturer instructions and using 500 ng of RNA as input material. qPCR was then performed for at least three biological replicates using Biozym Blue S'Green qPCR Mix Separate ROX (No. 331416XL) on a QuantStudio 7 Flex Real-Time PCR System from Applied Biosystems (primers listed in Supplementary Table 1). Pax3 fold-change was calculated from the differences between cycle thresholds (ΔCt) using Gapdh as housekeeping gene ($2^{-\Delta Ct}$). ΔΔCt was then calculated using DelBs mean as a reference value. Analysis of variance (ANOVA) testing has been performed to calculate group P value and two-sided Student's t-test for pairwise comparison.

**ChIP–seq.** Mouse embryonic fibroblast (MEF)-depleted mutant mESCs ($5 \times 10^6$) were washed twice with 1× PBS, dissociated with 1 ml of trypsin and centrifuged for 5 min at 114g at room temperature. The cell pellet was resuspended in 11.7 ml of 10% FCS and then fixed by adding 325 μl of 37% formaldehyde (Sigma-Aldrich) (final 1% formaldehyde) and incubated for 10 min at room temperature, while rotating. To stop the fixation process, the reaction was quenched on ice by adding 1 ml of 1.425 M glycine. Nuclei extraction was performed by adding 5 ml of ice-cold lysis buffer (10 mM Tris-HCl pH 7.5, 10 mM NaCl, 5 mM MgCl₂, 0.1 mM EGTA, 1× Protease Inhibitor (Roche Ref. 5892791001) in Milli-Q Water). Extracted nuclei were then collected by centrifugation at 460g for 5 min at 4 °C, washed with 1× PBS, snap-frozen and stored at −80 °C or further processed using iDeal ChIP for Transcriptional Factors Kit (Diagenode, Cat. No. C01010055). Briefly, cell nuclei were resuspended in 300 μl of Shearing Buffer and chromatin was sheared using Diagenode Bioruptor, to achieve a fragment size ranging from 200 base pairs (bp) to 500 bp. Immunoprecipitation was done using 15–20 μg of DNA and 1 μg of CTCF antibody (Diagenode, C15410210) and all steps were performed following the manufacturer instructions. ChIP–seq libraries were prepared using the NEBNext Ultra II Library Prep Kit for Illumina. Input material ranged from 500 pg to 15 ng of immunoprecipitated DNA and was processed according to the kit guidelines (NEBNext End Prep, Adaptor Ligation, PCR enrichment of Adaptor-Ligated DNA using NEBNext Multiplex Oligos for Illumina). Clean up and size selection were performed with AMPure beads (NEB). The library was sequenced with 30 million single-end reads of 75 nucleotides on a HiSeq4000 or NovaSeq platform.

**ChIPmentation.** Tagmentation-based chromatin immunoprecipitation (ChIPmentation) experiments were performed following published protocols[46]. Distal limb buds from E11.5 homozygous mouse mutants (male, Bl6/129Sv5) were microdissected in 1× PBS. The whole tissue was fixed with 1% PFA (in 10% FCS/PBS) for 10 min at room temperature while tumbling. The reaction was quenched on ice by adding glycine (125 mM final concentration) and the samples were centrifuged at 400g for 8 min at 4 °C. Later, the tissue was washed two times with cold 1× PBS, snap-frozen in liquid nitrogen and stored at −80 °C. The tissues were thawed on ice and resuspended in Chromatin Prep Buffer (High Sensitivity Chromatin Preparation kit, Cat. No. 53046 from Active Motif) with 1× Protease Inhibitor (Roche); nuclei were released using a Dounce homogenizer with a tight pistil (Active Motif Ref. 40415) and centrifuged at 4 °C at 1,250g for 5 min. Nuclei were then resuspended in cold Sonication Buffer (0.25% SDS, 10 mM Tris-HCl pH 8.0, 2 mM EDTA, 1× Protease Inhibitor) and pipetted to facilitate nuclei disruption. The chromatin was sonicated using a Diagenode Bioruptor (12 cycles of 20 s on, 30 s off) to achieve a fragment size ranging from 200 bp to 500 bp. To precipitate the debris, the chromatin was centrifuged at 15,871g for 10 min at 4 °C. Later, the sonicated chromatin was incubated overnight with pre-washed A Dynabeads (Thermo Fisher, Cat. No. 10001D) previously blocked with 0.1% BSA and 4 μg of anti-RAD21 antibody (Ab992). The following day, the beads were washed two times with RIPA-LS (10 mM Tris-HCl pH 8.0, 1 mM EDTA pH 8.0, 140 mM NaCl, 0.1% SDS, 0.1% Na-Deoxycolate, 1% Triton X-100, 1× Protease Inhibitor), two times with RIPA-HS (10 mM Tris-HCl pH 8.0, 1 mM EDTA pH 8.0, 500 mM NaCl, 0.1% SDS, 0.1% Na-Deoxycolate, 1% Triton X-100, 1× Protease Inhibitor), two times with RIPA-LiCl (10 mM Tris-HCl pH 8.0, 1 mM EDTA pH 8.0, 250 mM LiCl, 0.1% SDS, 0.5% Na-Deoxycolate, 1% Triton X-100, 1× Protease Inhibitor), two times with 10 mM Tris-HCl pH 8.0 and finally resuspended and incubated in the Tagmentation Solution (0.25% Tagmentation Buffer, 2 mM Tn5 from Illumina 20034197) for 2 min at 37 °C. The tagmentation reaction was stopped on ice by adding cold RIPA-LS. Later, beads were washed two times in RIPA-LS, two times in 1× Tris HCl-EDTA and finally resuspended in ChIP Elution Buffer (10 mM Tris-HCl pH 8.0, 5 mM EDTA pH 8.0, 300 mM NaCl, 0.4% SDS). Samples were de-crosslinked overnight and purified the day after using AmPure Beads (NEB). Libraries were prepared using KAPA HiFi HotStart ReadyMix (Roche) and indexed primers from ref. [46]. Libraries were purified using AmPure Beads (NEB) and sequenced with 30 million single-end reads of 75 nucleotides, on a NextSeq500 platform. All experiments were performed in duplicates, except for ΔR1 + F2 and R3-only mutants.

**cHi-C.** E11.5 mouse distal limb buds from homozygous mutants (male, Bl6/129Sv5) were microdissected in 1× PBS, resuspended and incubated in 1 ml of pre-warmed trypsin for 5–10 min at 37 °C. Trypsin was blocked by adding 5 ml of 10% FCS/PBS. The tissue was further dissociated to make a single-cell suspension by using a 40-μm cell-strainer (Product No. 352340) and finally centrifuged at 114g for 5 min at room temperature. The pellet was then resuspended in a 2% PFA (in 10% FCS/PBS) fixation solution and incubated at room temperature for 10 min while tumbling. To stop the fixation process, the reaction was quenched on ice by adding glycine (final concentration 125 mM) and centrifuged at 400g for 8 min at 4 °C. Nuclei extraction was performed by adding 1.5 ml of ice-cold lysis buffer (50 mM Tris-HCl pH 7.5, 150 mM NaCl, 5 mM EDTA, 0.5% NP-40, 1.15% Triton X-100, 25× Protease Inhibitor in Milli-Q water). Extracted nuclei were then collected by centrifugation at 750g for 5 min at 4 °C, washed with 1× PBS, snap-frozen and stored at −80 °C. The Chromosome Conformation Capture (3C) library was achieved by a DpnII digestion, a re-ligation of the digested fragments, de-crosslinking and DNA purification, and further processed using SureSelectXT Target Enrichment System for the Illumina Platform (Agilent Technology). Then, 200 ng to 3 μg of input material was sheared using a Covaris Sonicator and the following parameters: duty cycle: 10%; intensity: 5; cycles per burst: 200; time: 6 min; temperature: 4 °C. Sheared DNA was then processed following the kit guidelines (end repair, dA-tailing, adaptor ligation, PCR enrichment of adaptor-ligated DNA, DNA purification, hybridization and capture). The hybridization was performed using SureSelectXT Custom RNA probes library (Cat. No. 5190-4836) designed on the genomic region mm9 chr1:71,000,000–81,000,000. The capture was performed using Streptavidin-Coated Beads (Invitrogen). PCR enrichment and sample indexing were done following Agilent instructions. Capture libraries were sequenced with 400 million 75–100-bp paired-end reads on HiSeq4000 or NovaSeq platforms.

**cHi-C analysis.** Paired-end reads from all the cHi-C experiments were aligned using bwa mem local aligner[47] to a custom reference genome encompassing the captured region (chr1:71–81 megabases of the mm9 assembly) with the region corresponding to the baseline DelBs mutation deleted (chr1:76,838,978–77,858,974). There was one exception, the ΔF-all-Inv mutant cHi-C, in which a different version of the genome was used to account for the inverted coordinates (chr1:77,861,422–78,062,382). The rest of the chromosomes, including the remaining chr1, were kept in the custom reference genome to be able to distinguish not-uniquely mapped reads. Then, following the 4D Nucleome consortium recommendations, the resulting bam files were parsed with the pairtools suite (https://github.com/mirnylab/pairtools) to produce 4DN format files containing pairwise interactions. Briefly, bam files were parsed using pairtools parse. Then, not-uniquely mapped reads were filtered out using pairtools select (selecting UU, UR and RU pairs). Subsequently, pairs of reads were sorted and duplicated

pairs were removed using pairtools sort and pairtools dedup, respectively. Finally, dangling-ends were filtered out using a custom Python script available in the gitlab repository. Filtered 4DN-formatted pairs of interactions were then used to construct Knight–Ruiz (KR)-normalized Hi-C matrices in hic format with Juicer[48]. The hic files were further visualized and analyzed with FAN-C[49] and custom Python code also available in our gitlab repository. Briefly, insulation scores, boundaries and boundary scores were calculated as described elsewhere[32] using the dedicated FAN-C functions through the FAN-C API. Subtraction matrices were calculated as described elsewhere[28] with minor modifications. Briefly, first the coverage of the matrices to be subtracted was equalized, dividing by the total number of reads. Then, the two matrices were subtracted element-wise and each value of the subtraction was converted to a z-score taking into account the rest of the values belonging to the same sub-diagonal (corresponding to interactions happening at equivalent genomic distances). Virtual 4C tracks were visualized and quantified using custom Python and R scripts (available). Wild-type, *DelBs* and *DelB* cHi-C raw reads were downloaded from the Gene Expression Omnibus (GEO) (GSE92291)[28]. cHi-C experiments were performed in single replicates for each of the mutants following the rationale in ref. [28]. As an internal control, we compared the results from all six experiments for regions outside of the region of interest (excluding the coordinates from 4,300,000 to 6,200,000 of the described custom genome). Pearson correlation coefficients of pairwise comparisons were >0.89.

**ChIP–seq analysis.** Single-end reads were mapped to the same custom reference genomes as specified in the cHi-C section using Bowtie[50] (flags -m1 -S --chunkmbs 500). Reads aligning to more than one location were filtered out by bowtie due to the -m1 flag. Resulting alignments in SAM format were then converted to bam, sorted and deduplicated using Samtools (https://github.com/samtools/samtools). Then, deduplicated bam files were converted to bed including a 300-bp extension and subsequently bedGraph files containing the genome coverage were computed using BEDTools[51] (bamToBed and genomeCoverageBed respectively). Finally, bedGraph files were converted to bigWig files for visualization in UCSC genome browser (https://genome.ucsc.edu) using UCSC KentUtils (https://github.com/ucscGenomeBrowser/kent).

**ChIPmentation analysis.** Single-end reads from ChIP–seq sequencing were aligned using bowtie-1 to the same custom reference genome described for cHi-C (bowtie -m1 -t -S --chunkmbs 500). Reads aligning to two or more loci were discarded. Duplicated reads were removed using samtools rmdup and normalized coverage tracks in bigwig format were generated using deepTools BamCoverage after centering the read locations and extending the signal 300 bp (bamCoverage -e 300 --centerReads --normalizeUsing RPGC --effectiveGenomeSize 2620345972). Bigwig files were visualized in the UCSC genome browser (https://genome.ucsc.edu/). A Python wrapper performing the analysis from fastq files to bigwigs is available in the gitlab repository.

**Hi-C analysis.** *Data retrieval.* Already processed hic files[48] from high-resolution Hi-C datasets in mESCs, neural progenitors and cortical neurons[26] were obtained from the Juicebox repository (see index in https://hicfiles.tc4ga.com/juicebox.properties). CTCF ChIP–seq datasets from matching cell types were downloaded from GEO (see GSE96107). Hi-C data from mouse embryonic proximal and distal forelimbs[25] were also downloaded from GEO (see GSE101715) in validPairs format and subsequently converted to hic files using Juicer. Matching CTCF ChIP–seq data were obtained from GSE101714.

*Boundary analysis.* Insulation scores, boundaries and boundary scores[32] were calculated with FAN-C[49] using KR-normalized matrices at 25-kb resolution with a window size parameter of 250 kb. Boundaries located in the vicinity (±125 kb) of extremely low mappable regions were filtered out. Low-mappability regions were defined using a Gaussian mixture model on the marginal counts of the raw Hi-C matrices (further details and masked regions available in the gitlab repository). CBSs were predicted using CTCF peaks from matching ChIP–seq datasets, and CBS orientation was inferred using FIMO[52] (using the flags --bfile --motif-- --max-stored-scores 1000000 and the CTCF PWM from JASPAR, background estimated using MEME fasta-get-Markov utility). The highest scoring motif from each peak was retained for further analysis. For the mESC dataset, the total number of CBSs and the total number of divergent CBS pairs was then calculated for each boundary including a 100-kb-long flanking region call using BEDTools[51].

*Loop analysis.* We calculated loops using CPU hiccups[48] with the flags (--m 512 -r 5000,10000,25000 -k KR -f .1,.1,.1 -p 4,2,1 -i 7,5,3 -t 0.02,1.5,1.75,2 -d 20000,20000,50000) in 5-kb,10-kb and 25-kb KR-normalized matrices for the mESC dataset[26]. Loop anchors were intersected with the CBS information obtained as described in the Boundary analysis section using BEDTools. Then, loop anchors were classified accordingly into convergent-only (loop anchors that display at least one CBS oriented in the direction of all the loops they are engaged with), nonconvergent (loop anchors that are engaged in at least one loop that is formed despite lacking any CBS oriented in that direction) and non-CTCF (loop

anchors that do not display any CBSs). To calculate *P* values for the differential association of each of the three categories to the formation of bidirectional loops we performed a chi-squared test with 2 degrees of freedom. Being significant, we performed pairwise post hoc chi-squared tests and reported the Benjamini–Hochberg-corrected *P* values. CTCF loops were subsequently classified into two categories according to the nature of their anchors: convergent (loops formed by anchors displaying convergently oriented CBSs) and nonconvergent (if not). Convergent loops were further subdivided into single-sided convergent (if both anchors only engage in loops in the same direction), double-sided convergent (if at least one of the anchors engages in a convergent loop in the opposite direction) and convergent-associated (if at least one of the anchors engages in a nonconvergent loop in the opposite direction). Nonconvergent loops were also subdivided into simply nonconvergent and nonconvergent-associated (if at least one of the anchors is engaged in a convergent loop in the opposite direction). Loop strengths were calculated for each set of loops as previously proposed[22] using the dedicated FAN-C function[49]. Hi-C signal aggregates over the different loop categories were also calculated using FAN-C and 10-kb matrices. To test significance for differences in loop strength and loop anchor distances we first performed a Kruskal–Wallis test taking into account the five different categories of loops, followed by pairwise post hoc Mann–Whitney *U* tests. For the association of each of the five categories to loop anchors we performed a chi-squared test with 4 degrees of freedom followed by post hoc pairwise chi-squared tests. Benjamini–Hochberg-corrected *P* values are reported after pairwise Mann–Whitney *U* and chi-squared tests. The whole set of exact *P* values obtained is available in the gitlab repository.

**Statistical analyses.** Statistical tests used are always indicated in the corresponding Methods sections and in figure legends. Generally, for continuous variables, Mann–Whitney *U*-test nonparametric tests are used throughout the paper. For the comparison of continuous variables over more than two groups, Kruskal–Wallis *P* values smaller than 0.05 are required to perform further pairwise Mann–Whitney *U*-test *P* values which are reported after Benjamini–Hochberg correction for multiple testing. For qPCR, parametric ANOVA and Student's *t*-tests are used. For contingency analyses, chi-squared statistics are used followed by pairwise chi-squared tests corrected with Benjamini–Hochberg.

**Bioinformatic analyses and graphics.** Most statistical analyses related to Hi-C and cHi-C analyses were performed with Python (v.3.7.10) using the pandas (v.1.2.3), numpy (v.1.20.1), scipy (v.1.6.1) ecosystem. Multiple testing correction was performed with statsmodel (v.0.12.2) and pairwise tests with scikit-posthocs (v.0.6.6). Genomic interval operations were performed with pybedtools (v.0.8.2). Basic plotting, including barplots, boxplots, histograms and so on, was performed with seaborn (v.0.11.1) and matplotlib (v.3.3.3). Virtual 4C plots from cHi-C data were created with ggplot2 (v.3.3.2) within the R environment (v.4.1.0; dplyr v.1.0.7, data.table v.1.14.0). Genomic snapshots were either created with FAN-C (v.0.9.17) or extracted from UCSC Genome Browser visualizations (https://genome.ucsc.edu/).

**Reporting summary.** Further information on research design is available in the Nature Research Reporting Summary linked to this article.

## Data availability
The cHi-C, ChIP–seq and ChIPmentation datasets generated in this study have been deposited in the Gene Expression Omnibus (GEO) under the accession code GSE169561. Wild-type, *DelBs* and *DelB* cHi-C data are from a previous study[28] and raw reads were downloaded from GEO GSE92291. Already processed hic files[48] from high-resolution Hi-C datasets in mouse embryonic stem cells, neural progenitors and cortical neurons[26] were obtained from the Juicebox repository (see index in https://hicfiles.tc4ga.com/juicebox.properties). CTCF and RNAPII ChIP–seq datasets from matching cell types were downloaded from GEO (see GSE96107 and GSE112806). Hi-C data from mouse embryonic proximal and distal forelimbs[25] were also downloaded from GEO (see GSE101715) in validPairs format and subsequently converted to hic files using Juicer. Matching CTCF ChIP–seq data were obtained from GSE101714. Data for TF binding motifs were obtained from the JASPAR database (http://jaspar.genereg.net/).

## Code availability
Custom code is available at https://gitlab.com/rdacemel/anania2021.

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

## Acknowledgements

We thank the sequencing core, transgenic unit and animal facilities of the Max Delbrück Centre for Molecular Medicine and Max Planck Institute for Molecular Genetics for technical assistance. We thank K. Macura, J. Fiedler, C. Schwichtenberg, K. Zill, C. Franke, C. Scholl, M. Altmann, G. Kussagk, S. Bomberg, S. Reissert-Oppermann and C. Westphal for their support with the transgenic work. We thank S. Ali and M. Hochradel for their support in the preparation of capture Hi-C libraries. We thank E. Rodríguez-Carballo for his technical advice with the RAD21 ChIPmentation experiments. We thank C. Paliou, M. Franke, M. A. Martí-Renom and members of the Lupiañez laboratory for their valuable input and comments on the manuscript. This research was supported by a grant from the Deutsche Forschungsgemeinschaft (grant no. GA2495/1-1) and by a Helmholtz ERC Recognition Award grant from the Helmholtz-Gemeinschaft (grant no. ERC-RA-0033). R.D.A. was supported by an EMBO Postdoctoral Fellowship (grant no. EMBO ALTF 537-2020).

## Author contributions

C.A., R.D.A. and D.G.L. conceived the study and designed the experiments. C.A. performed most experiments with the support of A.B. and F.M.R. R.D.A. performed bioinformatic analysis. C.A., R.D.A. and D.G.L. analyzed the data. J.J., R.K. and L.W. performed tetraploid aggregations. G.C. and N.B. performed in situ hybridizations. C.A., R.D.A. and D.G.L. wrote the manuscript with input from all authors.

## Funding

## Competing interests

The authors declare no competing interests.

## Additional information

**Extended data** is available for this paper at https://doi.org/10.1038/s41588-022-01117-9.

**Correspondence and requests for materials** should be addressed to Darío G. Lupiáñez.

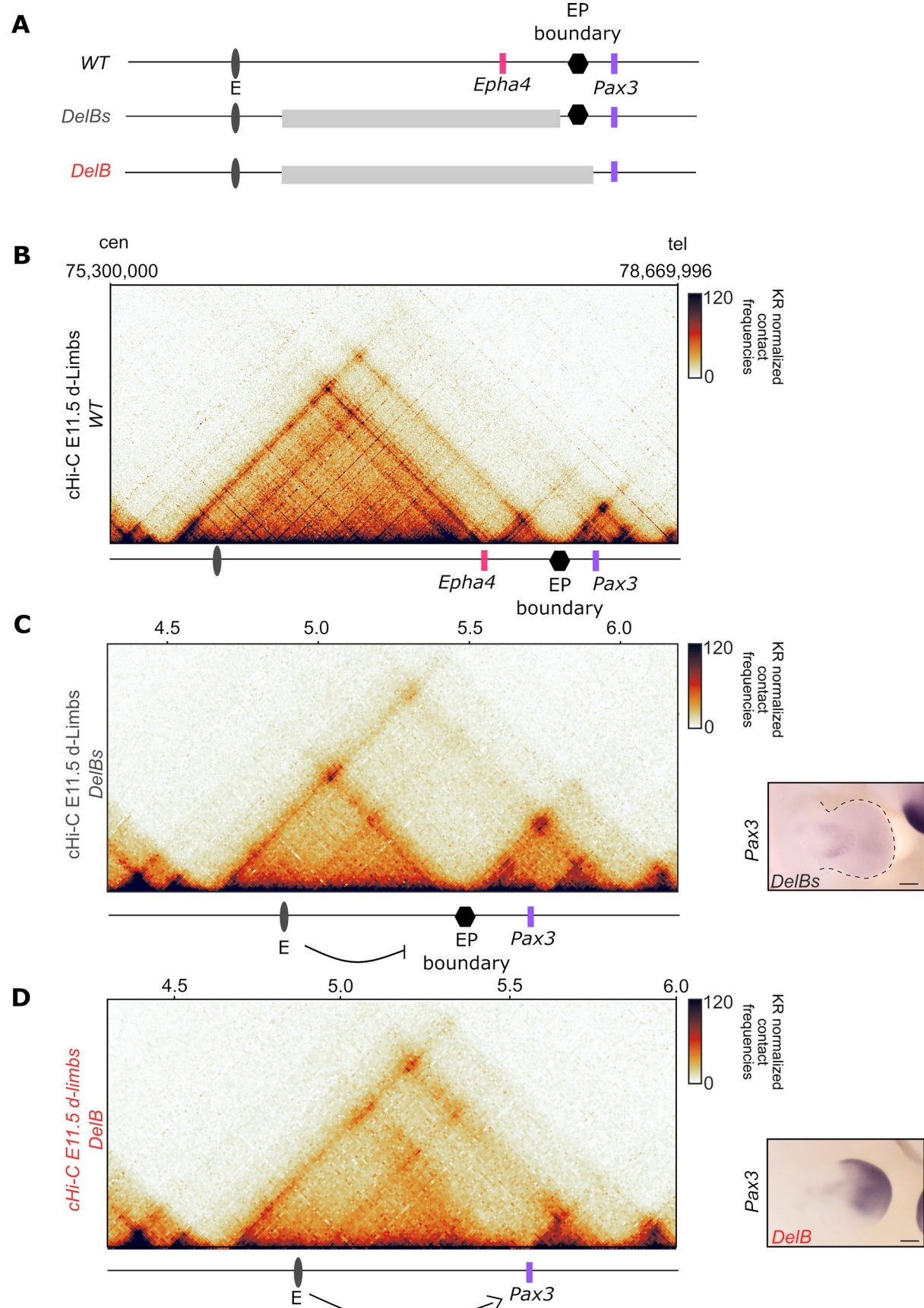

**Extended Data Fig. 1 | See next page for caption.**

**Extended Data Fig. 1 | Structural and molecular comparison between *Delbs* and *DelB* mutants. a**. Schematic shows *Epha4* enhancers (in dark gray), *Epha4* gene (in pink), *Pax3* gene (in purple), EP boundary (in black) and genomic rearrangements in the *DelBs* and *Delb* mutants (light gray rectangles). **b-d**. Capture Hi-C (cHi-C) maps in E11.5 limbs in WT (**B**) *DelBs* (**C**) and *DelB* (**D**) mutants (data from[28]). Genomic coordinates in C and D correspond to custom *DelBs* and *DelB* genomes for the captured region, respectively. *Pax3* WISH (right panel). Note how the presence of the EP boundary is sufficient to block the functional interaction between the *Epha4* enhancers and the *Pax3* gene, thus preventing its misexpression. Cen, centromeric. Tel, telomeric. d-Limbs, distal limbs. Scale bars, 250 μm.

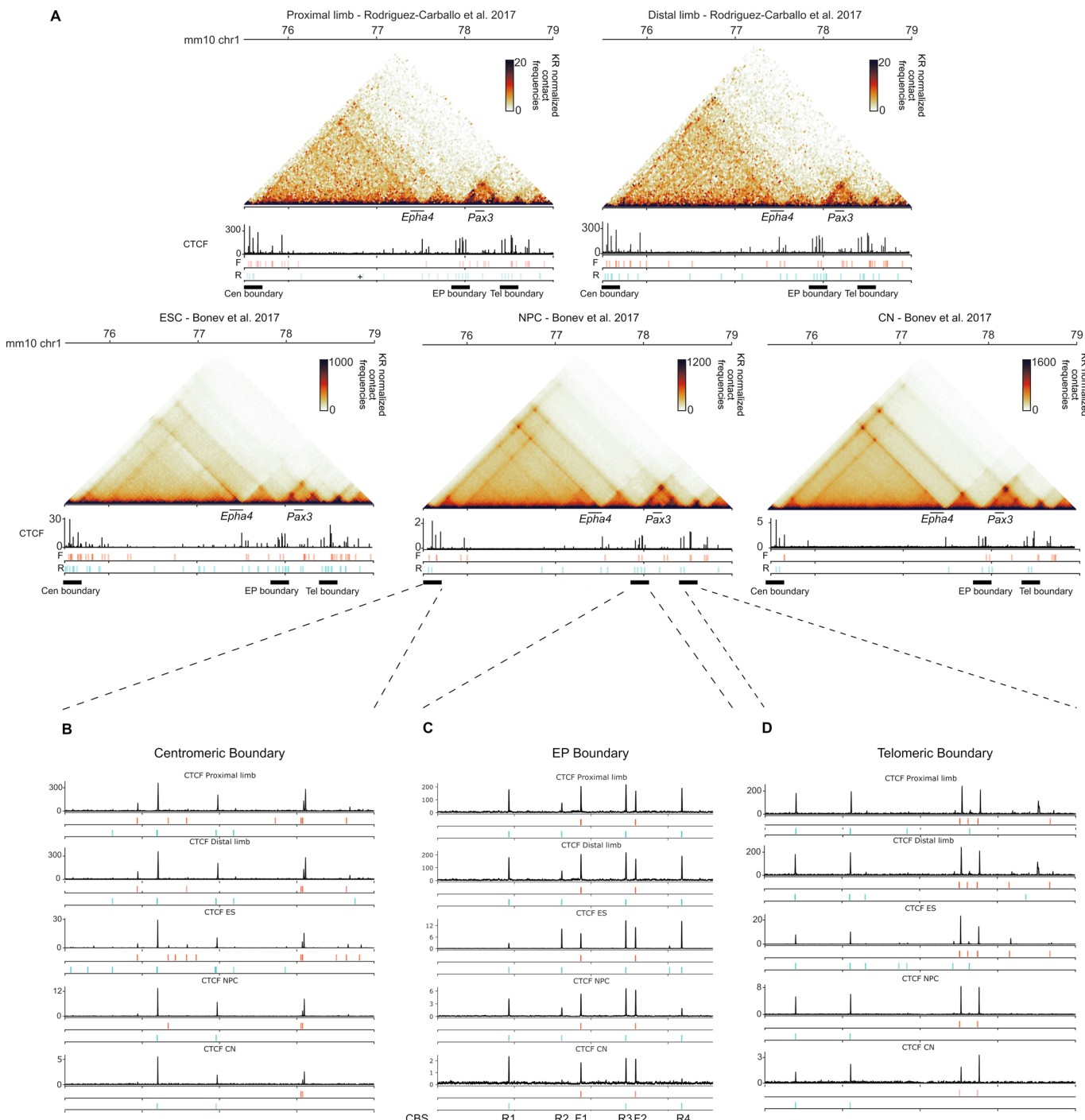

**Extended Data Fig. 2 | *Epha4-Pax3* (EP) boundary is a constitutive boundary with stable binding of divergently oriented CTCF. a**. Hi-C maps at 25 kb resolution and CTCF ChIP-seq tracks around the EP boundary locus from 5 different sources are shown. CTCF motifs inside CTCF peaks in a forward (F) or reverse (R) orientation are depicted below the ChIP-seq track in red and blue respectively. Motifs were calculated using FIMO[52] (see methods). The first two datasets are proximal and distal embryonic forelimbs respectively[25]. The other three datasets correspond to the high-resolution datasets in mESC, neural progenitor cells and cortical neurons from[26]. **b–d**. Close-ups of the corresponding CTCF ChIP-seq experiments in the centromeric (Cen), *Epha4-Pax3* (EP) and telomeric (Tel) boundaries respectively.

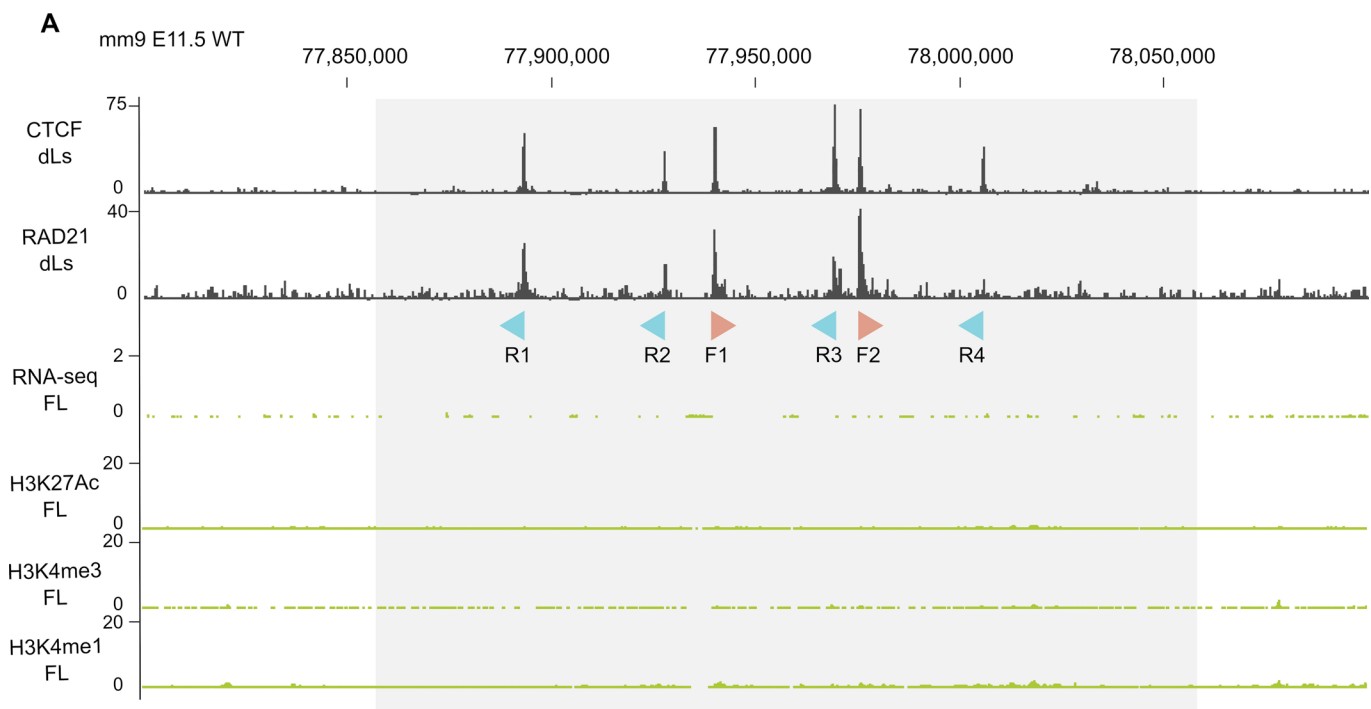

**Extended Data Fig. 3 | Epigenetic landscape at the *Epha4-Pax3* locus. a**. Genome browser tracks showing CTCF and RAD21 ChIP-seq from E11.5 wild-type distal limbs (dLs), and H3K4me3, H3K27Ac, H3K4me1 ChIP-seq and RNA-seq of E11.5 wild-type mouse forelimbs (FL) (data from[25]). Note how the EP boundary is occupied by CTCF and RAD21, but shows no presence of the other histone marks nor of active transcription. EP boundary is indicated by the gray box. Light blue and orange arrowheads represent reverse (R) and forward (F) oriented CBS, respectively.

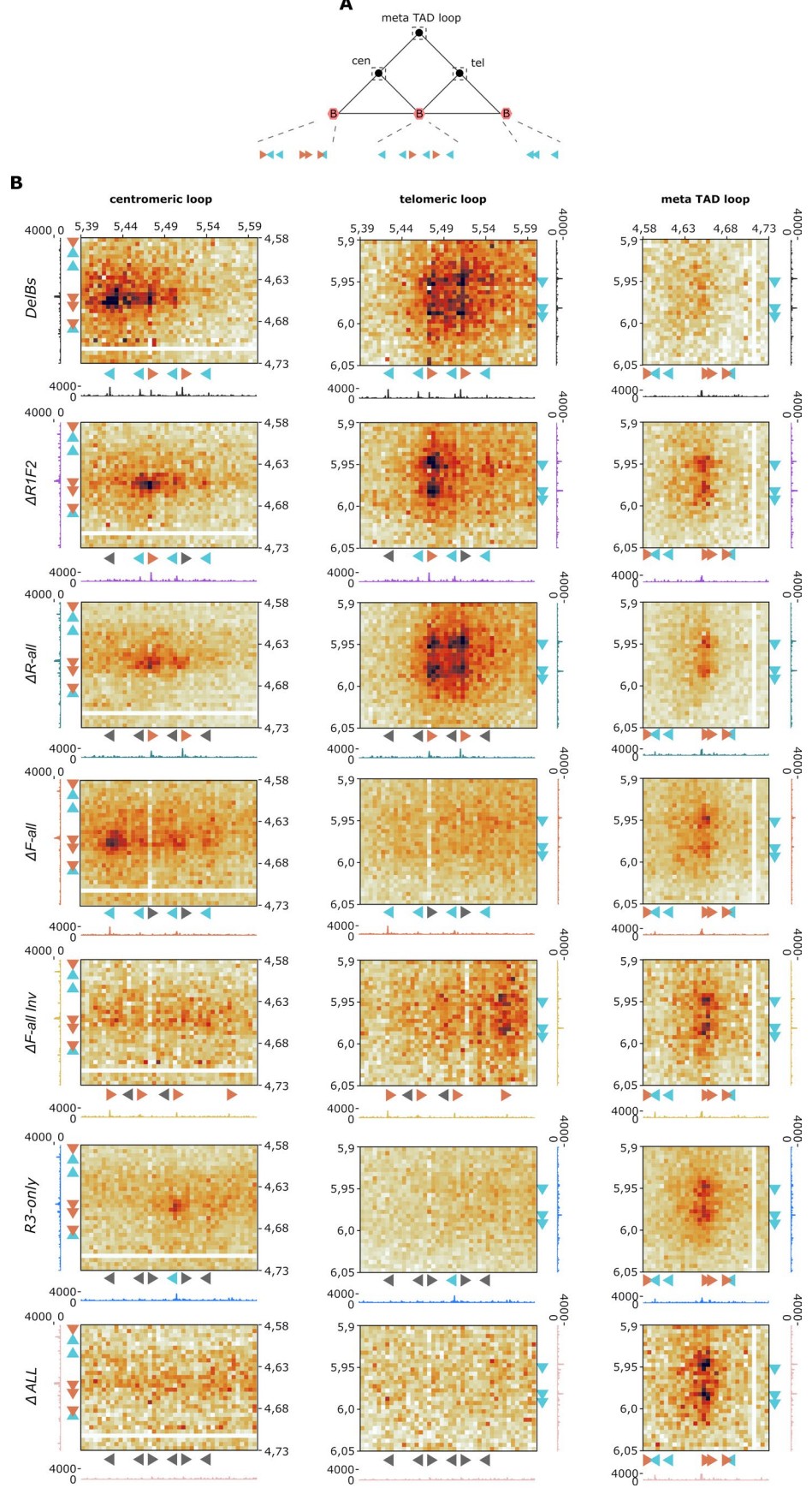

**Extended Data Fig. 4 | See next page for caption.**

**Extended Data Fig. 4 | Nonconvergent centromeric loops mediated by the F1 and F2 CBS. a**. Schematic of the 3D configuration of the locus. The *Epha4* TAD (left), formed by the centromeric loop (cen), and the *Pax3* TAD, formed by the telomeric loop (tel), are both highlighted by the dashed squares. The centromeric and telomeric loops are anchored on one side by the EP boundary, and on the other side by the centromeric and telomeric boundaries, respectively. In the dashed square, on top, is highlighted a meta-TAD loop, anchored by the centromeric and telomeric boundaries. The boundaries are composed of clusters of CBS, depicted by orange and blue arrowheads (forward and reverse oriented, respectively. **b**. Close-up of the cHi-C interaction matrices showing the centromeric, telomeric and meta-TAD loops established by the remaining CBS of the EP boundary and the centromeric and telomeric counterparts, in the different mutants. Below and beside each close-up, RAD21 ChIP-seq tracks for each boundary involved in the loop. Deleted CBS are indicated in gray. The coordinates shown are 4.55 Mb to 4.75 Mb for the centromeric loops and 5.8 Mb to 6.1 Mb for the telomeric. Genomic coordinates correspond to a custom *DelBs* genome for the captured region.

**A**

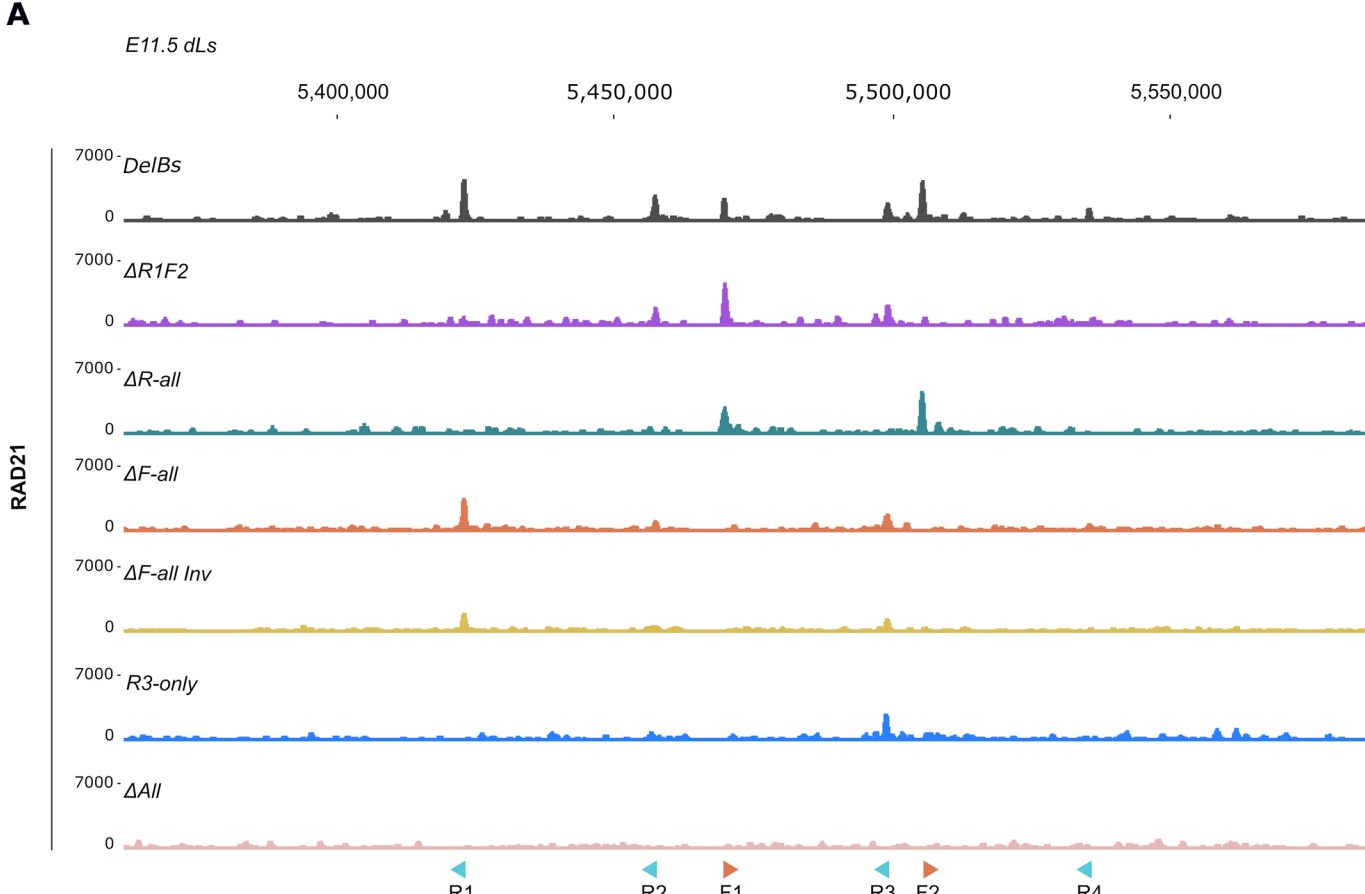

**Extended Data Fig. 5 | RAD21 ChIP-seq in CBS mouse mutants. a**. RAD21 ChIP-seq experiments performed in distal limbs (dLs) of E11.5 mouse mutants. ChIP-seq tracks show absence of RAD21 occupancy at the deleted CBS. The locations of the wild-type CBS are depicted with orange and blue arrowheads (forwards and reverse, respectively).

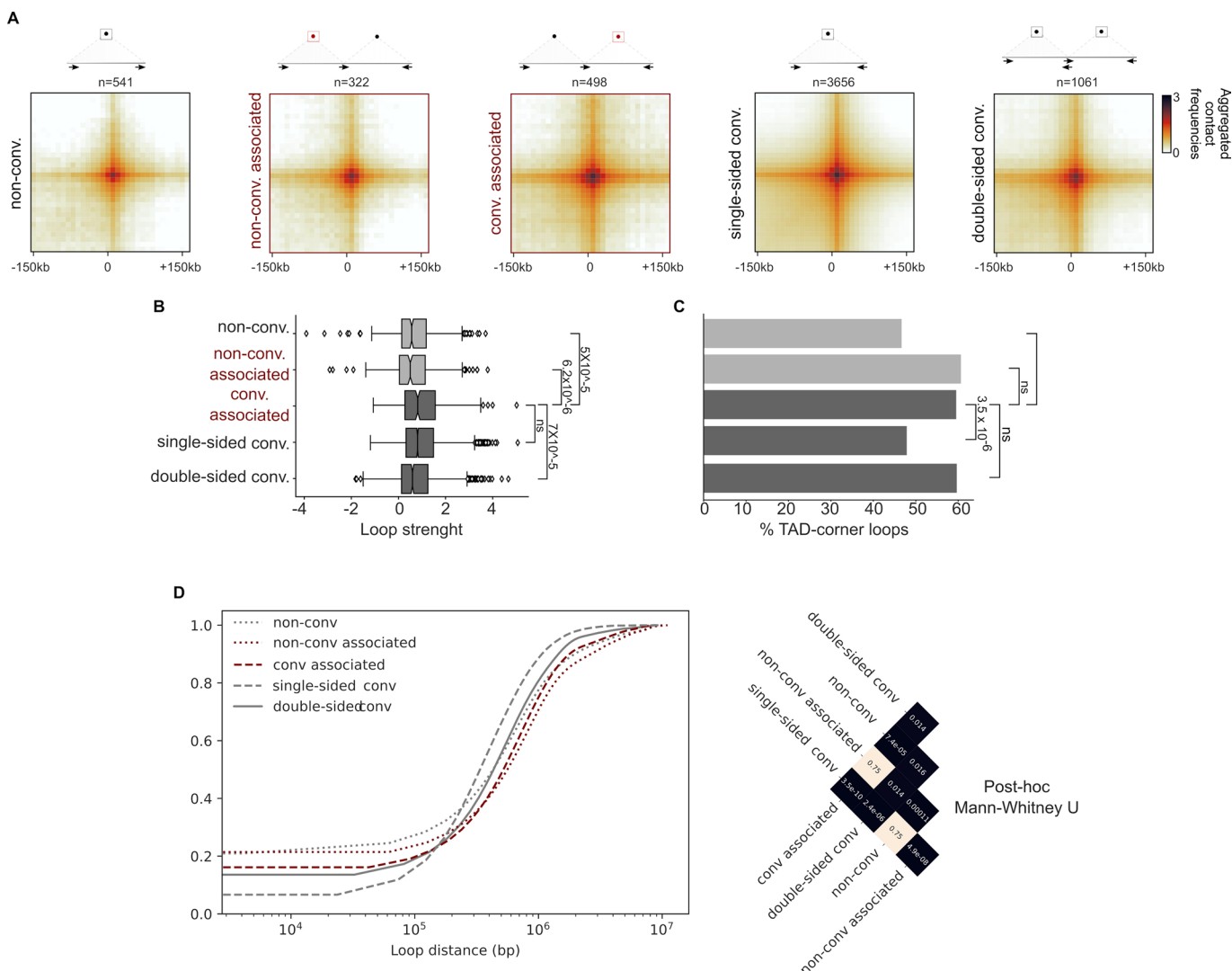

**Extended Data Fig. 6 | Paired convergent/nonconvergent loops display longer distances between anchors and more association to TAD-corner loops than unidirectional convergent loops. a**. Schematic (above) and loop aggregate plots (below) for all possible loop categories depending on the anchors: convergent (conv.) and nonconvergent (non-conv.). Convergent loops can belong to the conv. associated if they share an anchor with a nonconvergent loop in the opposite direction (n = 498). If not, they can be either single-sided (if both their anchors establish loops in a single orientation, n = 3656) or double-sided (n = 1061). Nonconvergent loops are further subdivided in non-conv. associated if they share an anchor with a convergent loop in the opposite direction (n = 322) or simply non-conv if they do not (n = 541). Non-conv. associated and conv. associated loops are depicted in red because these categories associate to each other. **b**. Boxplots show the loop strength for the loop categories described in A. The boxes in the boxplots indicate the median and the first and third quartiles (Q1 and Q3). Whiskers extend to the last observation within 1.5 times the interquartile range below and above Q1 and Q3 respectively. The rest of observations, including maxima and minima, are shown as outliers. Significant two-sided and Benjamini-Hochberg corrected Mann-Whitney U p-values are shown in the appropriate comparisons between conv. associated loops and the rest of categories. **c**. Barplots show the percentage of loops associated with putative TAD-corner loops for each of the categories shown in A and B. Significant differences between convergent associated loops and the rest of categories are highlighted with Benjamini-Hochberg corrected pairwise χ2 p-values when appropriate. **d**. Left: cumulative distribution of loop distances in the previous categories of loops. Right: Benjamini-Hochberg corrected two-sided Mann-Whitney U p-values are shown.

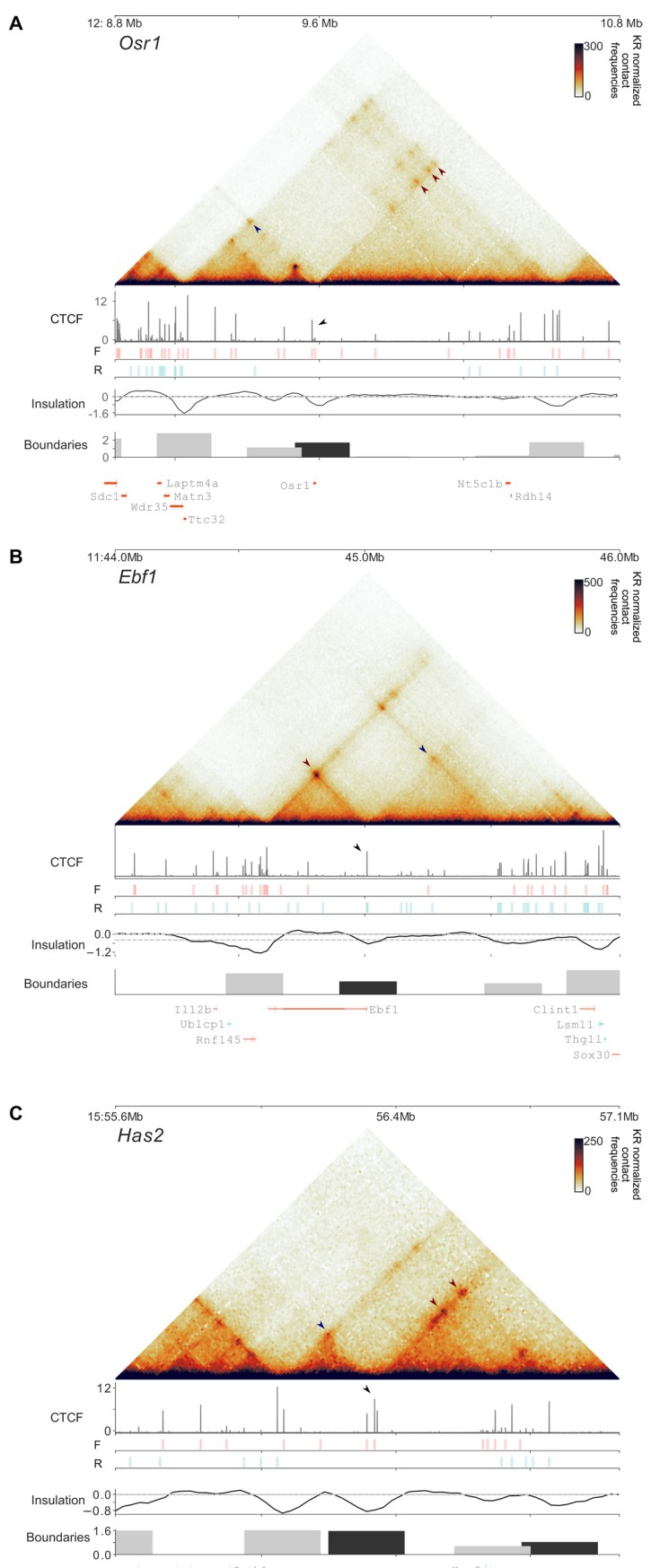

**Extended Data Fig. 7 | See next page for caption.**

**Extended Data Fig. 7 | Loop anchors with strong single-oriented CTCF binding can constitute the source of weaker loops in the nonconvergent direction. a-c**. Three different examples of developmental gene loci with loop anchors that display unidirectional CBS and are engaged in both convergent and nonconvergent loops. Hi-C interaction matrices and CTCF ChIP-seqs are from the mESC dataset in[26]. CBS orientations are displayed in red and blue for positive and negative strands respectively and were calculated using FIMO[52] (see methods). Dark red and dark blue arrowheads indicate convergent and associated nonconvergent loops respectively. Insulation scores, boundaries and boundary scores were calculated with FAN-C[49] (see methods). Black boundary bars depict boundaries that do not contain divergent CBS pairs in their vicinity.

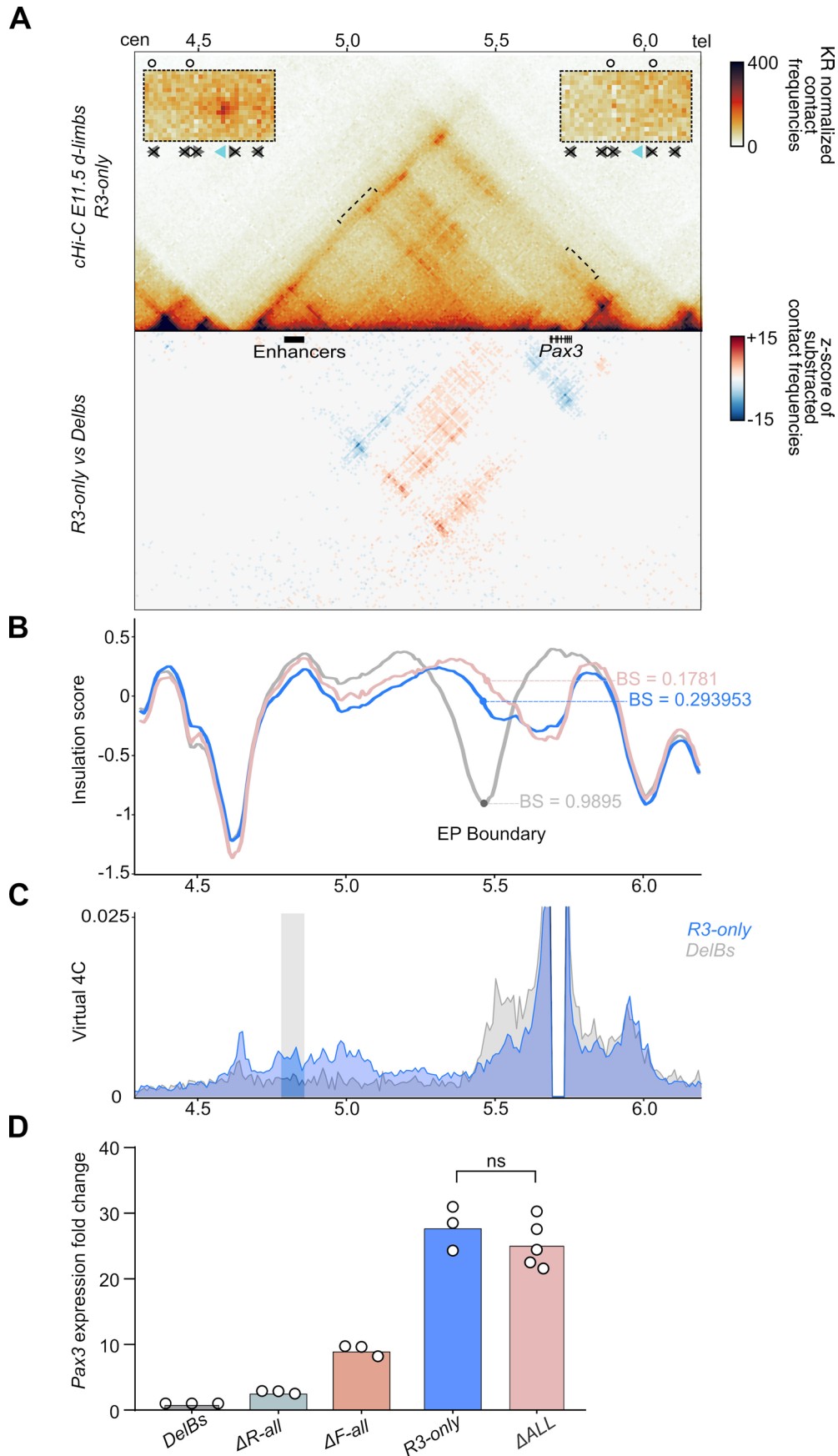

**Extended Data Fig. 8 | See next page for caption.**

**Extended Data Fig. 8 | TAD fusion in *R3-only* mutants. a**. cHi-C maps from E11.5 mutant distal limbs at 10 kb resolution (top). Data mapped on custom genome containing the *DelBs* deletion. Insets represent a magnification (5 kb resolution) of the centromeric (left) and telomeric (right) loops. Highlighted by brackets on the map. Lost chromatin loops represented by empty dots. Subtraction maps (bottom) showing gain (red) or loss (blue) of interactions in mutants compared to *DelBs*. **b**. Insulation score values. Lines represent mutants. Dots represent the local minima of the insulation score at EP boundary for each mutant, also measured as boundary score (BS). **c**. Virtual 4C profiles with *Pax3* promoter as a viewpoint for the genomic region displayed in panel A. Light gray rectangle highlights *Epha4* enhancer region. Note increased interactions between *Pax3* promoter and *Epha4* enhancer in *R3-only* (blue) compared to *DelBs* mutant (gray). **d**. *Pax3* qPCR analysis in E11.5 limb buds from CBS mutants. Bars represent the mean and white dots represent individual replicates. Values normalized against *DelBs* mutant (ΔΔCt). Note no difference in *Pax3* misexpression between *R3-only* and *ΔALL* (two-sided T-test p-value ns: non significant).

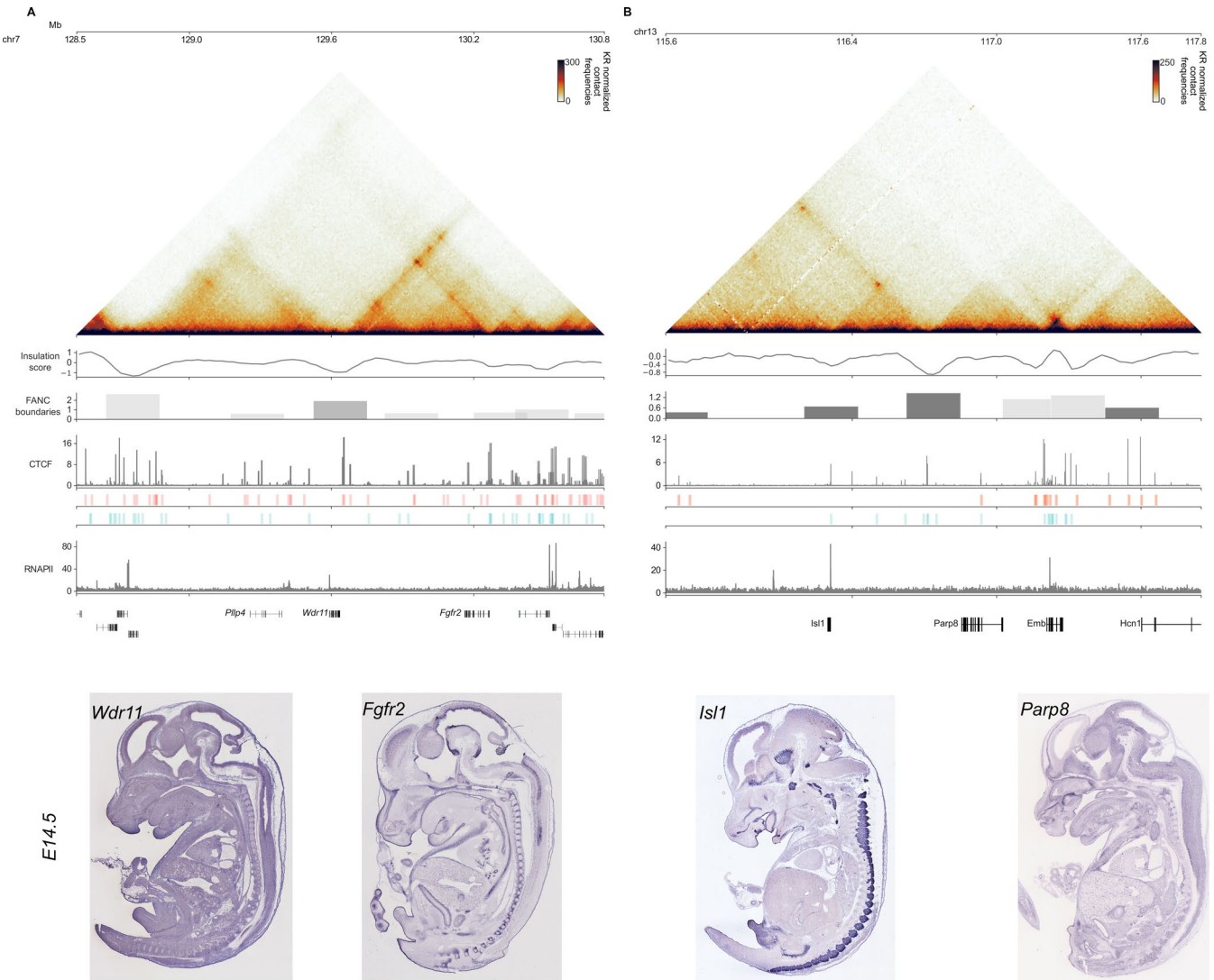

**Extended Data Fig. 9 | Boundary elements containing CBS in a single orientation can achieve comparable levels of insulation compared to boundaries containing divergent CBS.** Two different examples of developmental loci, **(a)** *Fgfr2* and **(b)** *Isl1*, where boundary elements containing single-oriented CBS achieve boundary scores higher than one. Above, Hi-C and CTCF ChIP-seq experiments from mESC are shown[26]. CBS are depicted in red or blue for forward and reverse orientation respectively. Insulation scores, boundaries and boundary elements are calculated with FAN-C[49] (see Methods). RNAPII ChIP-seq experiments in mESC (GSE112806) do not show a particular enrichment in either of the boundaries. Below, E14.5 WISH from representative genes at either side of the boundary do not suggest co-regulation (obtained from GenePaint.org).

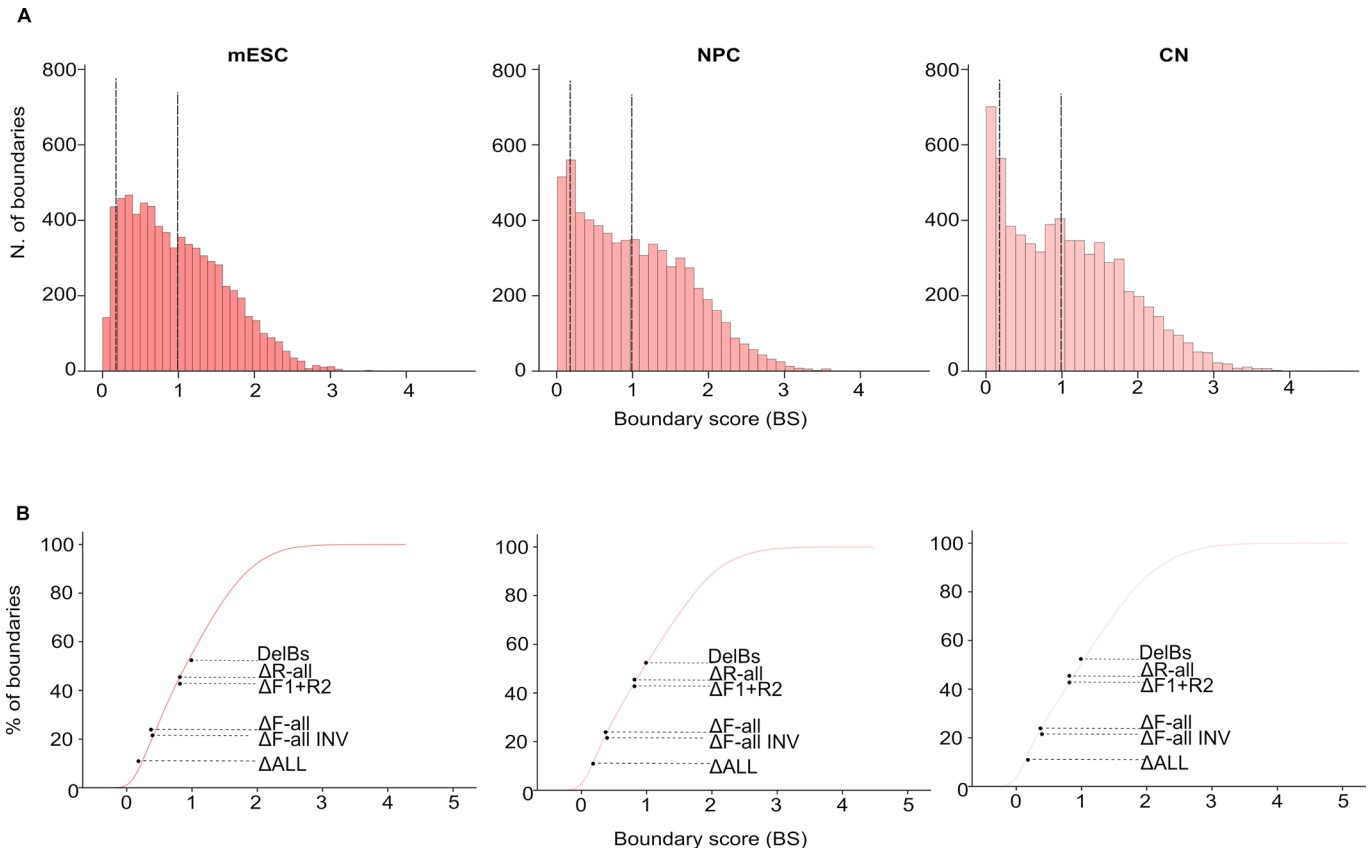

**Extended Data Fig. 10 | The boundary score (BS) of 40% of boundaries genome-wide could potentially allow regulatory inter-boundary interactions.**
**a**. Histogram representing the distribution of Boundary Scores genome-wide calculated from the mESC (left), neural progenitor cells (center) and cortical neurons (right) Hi-C datasets[26] Many of them fall within the range of boundary scores of the EP boundary in our mutant series (demarcated by the vertical dashed lines). **b**. Cumulative distribution of the boundary scores from A. The boundary scores of the EP boundaries in each of our mutants is highlighted.

# Reporting Summary

Nature Research wishes to improve the reproducibility of the work that we publish. This form provides structure for consistency and transparency in reporting. For further information on Nature Research policies, see our Editorial Policies and the Editorial Policy Checklist.

## Statistics

For all statistical analyses, confirm that the following items are present in the figure legend, table legend, main text, or Methods section.

| n/a | Confirmed | |
|---|---|---|
| ☐ | ☒ | The exact sample size (*n*) for each experimental group/condition, given as a discrete number and unit of measurement |
| ☒ | ☐ | A statement on whether measurements were taken from distinct samples or whether the same sample was measured repeatedly |
| ☐ | ☒ | The statistical test(s) used AND whether they are one- or two-sided<br>*Only common tests should be described solely by name; describe more complex techniques in the Methods section.* |
| ☒ | ☐ | A description of all covariates tested |
| ☐ | ☒ | A description of any assumptions or corrections, such as tests of normality and adjustment for multiple comparisons |
| ☐ | ☒ | A full description of the statistical parameters including central tendency (e.g. means) or other basic estimates (e.g. regression coefficient) AND variation (e.g. standard deviation) or associated estimates of uncertainty (e.g. confidence intervals) |
| ☐ | ☒ | For null hypothesis testing, the test statistic (e.g. *F*, *t*, *r*) with confidence intervals, effect sizes, degrees of freedom and *P* value noted<br>*Give P values as exact values whenever suitable.* |
| ☒ | ☐ | For Bayesian analysis, information on the choice of priors and Markov chain Monte Carlo settings |
| ☒ | ☐ | For hierarchical and complex designs, identification of the appropriate level for tests and full reporting of outcomes |
| ☐ | ☒ | Estimates of effect sizes (e.g. Cohen's *d*, Pearson's *r*), indicating how they were calculated |

*Our web collection on statistics for biologists contains articles on many of the points above.*

## Software and code

Policy information about availability of computer code

| Data collection | No software was used for data collection. |
|---|---|
| Data analysis | Custom code is available in a gitlab repository (https://gitlab.com/rdacemel/anania2021).<br><br>The following publicly available programs were used:<br>bwa (v0.7.17-r1188)<br>pairtools (v0.3.0)<br>juicertools (v1.22.01)<br>fanc (v0.9.17).<br>bowtie (v1.2.3)<br>samtools (v1.9)<br>bedtools (v2.29.2)<br>bedGraphToBigWig (kentUtils v4)<br>FIMO (MEME, v5.1.1)<br>deeptools (v3.5.1)<br><br>The following python==3.7.10 package versions were used:<br>pandas==1.2.3<br>numpy==1.20.1<br>pybedtools==0.8.2<br>seaborn==0.11.1<br>matplotlib==3.3.3<br>scikit-learn==0.24.1 |

For manuscripts utilizing custom algorithms or software that are central to the research but not yet described in published literature, software must be made available to editors/reviewers. We strongly encourage code deposition in a community repository (e.g. GitHub). See the Nature Research guidelines for submitting code & software for further information.

```
scipy==1.6.1
statsmodels==0.12.2
scikit-posthocs==0.6.6

The following R==4.1.0 packages versions were used:
dplyr==1.0.7
data.table==1.14.0
ggplot2==3.3.2
```

For manuscripts utilizing custom algorithms or software that are central to the research but not yet described in published literature, software must be made available to editors and reviewers. We strongly encourage code deposition in a community repository (e.g. GitHub). See the Nature Research guidelines for submitting code & software for further information.

## Data

Policy information about availability of data

All manuscripts must include a data availability statement. This statement should provide the following information, where applicable:

- Accession codes, unique identifiers, or web links for publicly available datasets
- A list of figures that have associated raw data
- A description of any restrictions on data availability

Sequencing data from cHi-C, ChIPmentation and ChIP-seq experiments is available in GEO (GSE169561).

# Field-specific reporting

Please select the one below that is the best fit for your research. If you are not sure, read the appropriate sections before making your selection.

☒ Life sciences ☐ Behavioural & social sciences ☐ Ecological, evolutionary & environmental sciences

For a reference copy of the document with all sections, see nature.com/documents/nr-reporting-summary-flat.pdf

# Life sciences study design

All studies must disclose on these points even when the disclosure is negative.

| | |
|---|---|
| Sample size | Sample size was at least 3 homozygous animals for qPCR and WISH. Phenotypical analysis on mutant embryos were performed for at least 4 homozygous animals. cHi-C experiments were performed in single-replicates using a pool of distal limbs from different embryos. These are commonly accepted sample sizes for these types of experiments (Franke et al., Nature, 2016; Bianco et al., Nature Genetics 2018; Rouco et al., Nature Communications, 2021). Rad21 ChIPmentation experiments were performed in duplicates as is the standard set by the ENCODE project for this type of experiments. CTCF ChIP-seq experiments were performed in one replicate since they were only used as a control to certify the total absence of CTCF binding upon CTCF binding site deletions in the different mutant cell lines |
| Data exclusions | Samples were excluded only according to the genotype which was analysed in control experiments. |
| Replication | At least 3 embryos with homozygous genotype were used to perform WISH experiments and they gave reproducible staining. At least 4 embryos were used to perform skeletal staining and phenotypical analysis giving reproducible results. Statistical analysis was performed on the measurements of the finger lengths which are stated in the figure legend of Main Fig. 6. Individual embryos datapoint for the finger lengths are presented in Main Fig. 6. At least 3 embryos were used to perform qPCR experiments, individual datapoints are presented in the figures. Statistical analysis was performed on the qPCR experiments which is stated in the manuscript and in the figure legends. Reproducibility of single replicate cHi-C experiments was found to be high outside from the two TADs of interest (proposed in Bianco et al., Nature Genetics 2018; Pearson > 0.89, see more details in Methods). Rad21 ChIPmentation experiments were performed in duplicates as is the standard set by the ENCODE project for this type of experiments with the exception of R3-only and R1+F2 genotypes due to difficulties obtaining enough embryonic material from these genetic backgrounds. CTCF ChIP-seq experiments were performed in one replicate since they were only used as a control to certify the total absence of CTCF binding upon CTCF binding site deletions in the different mutant cell lines |
| Randomization | Randomization is not relevant on this study because all the experiments performed had to take into account the genotype of the samples and there is no treatment involved nor additional covariates expected to introduce biases in development. |
| Blinding | The experiments were not performed blindly because the embryos generation and analysis required knowledge about their genotype and all comparisons were performed automatically using statistical software that is not influenced by the investigator. |

# Reporting for specific materials, systems and methods

We require information from authors about some types of materials, experimental systems and methods used in many studies. Here, indicate whether each material, system or method listed is relevant to your study. If you are not sure if a list item applies to your research, read the appropriate section before selecting a response.

## Materials & experimental systems

| n/a | Involved in the study |
|-----|----------------------|
| ☐ | ☒ Antibodies |
| ☐ | ☒ Eukaryotic cell lines |
| ☒ | ☐ Palaeontology and archaeology |
| ☐ | ☒ Animals and other organisms |
| ☒ | ☐ Human research participants |
| ☒ | ☐ Clinical data |
| ☒ | ☐ Dual use research of concern |

## Methods

| n/a | Involved in the study |
|-----|----------------------|
| ☐ | ☒ ChIP-seq |
| ☒ | ☐ Flow cytometry |
| ☒ | ☐ MRI-based neuroimaging |

# Antibodies

| | |
|---|---|
| Antibodies used | CTCF-Ab Diagenode Cat. No. C15410210 Lot. No. A2359-00234D (1ug per IP) also stated in Material and Methods<br>RAD21-Ab ABCAM Cat. No. ab992 Lot. N0. GR3310168 (4ug per IP) also stated in Material and Methods<br>Anti-Digoxigenin-AP, Fab fragments ROCHE Cat. No. 11093274910 (used 1:5000) also stated in Material and Methods |
| Validation | CTCF-Ab<br>Polyclonal ChIP-seq grade. Species reactivity: human and mice. Host: Rabbit. Applications: ChIP/ChIP-seq, ELISA, Western Blotting, Immunofluorescence. Validation by the manufacturer: determination of Ab titer by ELISA; validation in HeLa cells by ChIP-qPCR, ChIP-seq, Western Blot and Immunofluorescence.<br>RAD21-Ab<br>Polyclonal. Species reactivity: human and mice. Host: Rabbit. Applications: Immunoprecipitation and Western Blotting. Validation by the manufacturer: validation by Immunoprecipitation in Hep3B human cell lysate; validation by Western Blot in different mouse and human cell lysate.<br>Anti-Digoxigenin-AP, Fab fragments<br>Polyclonal anti-digoxigenin antibodies. Host: sheep. Applications: cDNA array, Colony/plaque hybridization, Dot blot, ELISA, Gel shift assay, Immunohistocytochemistry, In situ hybridization, Nonradioactive DNA sequencing blot, Northern blot, RNase protection assay, Southern blot, Western blot, Fluorescent in situ hybridization, Section in situ hybridization and whole mount in situ hybridization, Electrophoretic mobility shift assay. Validation by the manufacturer by Dot blot, ELISA, Immunohistocytochemistry, In situ hybridization, Southern blot and Western blot. |

# Eukaryotic cell lines

Policy information about cell lines

| | |
|---|---|
| Cell line source(s) | G4F1 (https://doi.org/10.1073/pnas.0609277104)<br>G4F1 /DelBs<br>G4F1 /DelB<br>G4F1 /ΔR1- ΔR2- ΔR3- ΔR4- ΔF1- ΔF2<br>G4F1 /ΔR1+F2<br>G4F1 /ΔR-all<br>G4F1 /ΔF-all<br>G4F1/ΔF-all-Inv<br>G4F1/ΔALL<br>G4F1/R3-only |
| Authentication | Cell-lines were not authenticated |
| Mycoplasma contamination | All cells were tested for mycoplasma contamination using Mycoalert detection kit (Lonza) and Mycoalert assay control set (Lonza) |
| Commonly misidentified lines<br>(See ICLAC register) | No commonly misidentified lines cell lines were used. |

# Animals and other organisms

Policy information about studies involving animals; ARRIVE guidelines recommended for reporting animal research

| | |
|---|---|
| Laboratory animals | Mus musculus, CD-1, female, various ages for donor and embryos retransfered by tetraploid aggregation and E11.5 and E17.5 embryos isolated to perform experimental analysis. |
| Wild animals | This study did not involve wild animals. |
| Field-collected samples | This study did not involve field-collected samples. |
| Ethics oversight | Mice were handled according to institutional guidelines under an experimentation license (G0111/17) approved by the Landesamt fuer Gesundheit und Soziales (Berlin, Germany) |

Note that full information on the approval of the study protocol must also be provided in the manuscript.

# ChIP-seq

## Data deposition

☒ Confirm that both raw and final processed data have been deposited in a public database such as GEO.

☒ Confirm that you have deposited or provided access to graph files (e.g. BED files) for the called peaks.

| | |
|---|---|
| **Data access links**<br>*May remain private before publication.* | Sequencing data from cHi-C, ChIPmentation and ChIP-seq experiments is available in GEO (GSE169561). |
| **Files in database submission** | ChIP_CTCF_mESC_DelBs.fq.gz ChIP_CTCF_mESC_DelBs.bw<br>ChIP_CTCF_mESC_F1.fq.gz ChIP_CTCF_mESC_F1.bw<br>ChIP_CTCF_mESC_F2.fq.gz ChIP_CTCF_mESC_F2.bw<br>ChIP_CTCF_mESC_F-ALL.fq.gz ChIP_CTCF_mESC_F-ALL.bw<br>ChIP_CTCF_mESC_F-ALL-INV.fq.gz ChIP_CTCF_mESC_F-ALL-INV.bw<br>ChIP_CTCF_mESC_R1.fq.gz ChIP_CTCF_mESC_R1.bw<br>ChIP_CTCF_mESC_R2.fq.gz ChIP_CTCF_mESC_R2.bw<br>ChIP_CTCF_mESC_R3.fq.gz ChIP_CTCF_mESC_R3.bw<br>ChIP_CTCF_mESC_R4.fq.gz ChIP_CTCF_mESC_R4.bw<br>ChIP_CTCF_mESC_R-ALL.fq.gz ChIP_CTCF_mESC_R-ALL.bw<br>ChIP_CTCF_mESC_ALL.fq.gz ChIP_CTCF_mESC_ALL.bw<br>ChIP_CTCF_DistalLimb_E11-5_WT_r1.fq.gz ChIP_CTCF_DistalLimb_E11-5_WT_r1.bw<br>ChIP_Rad21_DistalLimb_E11-5_ALL_r1.fq.gz ChIP_Rad21_DistalLimb_E11-5_ALL_r1.bw<br>ChIP_Rad21_DistalLimb_E11-5_ALL_r1_input.fq.gz ChIP_Rad21_DistalLimb_E11-5_ALL_r1_input.bw<br>ChIP_Rad21_DistalLimb_E11-5_ALL_r2.fq.gz ChIP_Rad21_DistalLimb_E11-5_ALL_r2.bw<br>ChIP_Rad21_DistalLimb_E11-5_ALL_r2_input.fq.gz ChIP_Rad21_DistalLimb_E11-5_ALL_r2_input.bw<br>ChIP_Rad21_DistalLimb_E11-5_DelBs_r1.fq.gz ChIP_Rad21_DistalLimb_E11-5_DelBs_r1.bw<br>ChIP_Rad21_DistalLimb_E11-5_DelBs_r2.fq.gz ChIP_Rad21_DistalLimb_E11-5_DelBs_r2.bw<br>ChIP_Rad21_DistalLimb_E11-5_DelBs_r2_input.fq.gz ChIP_Rad21_DistalLimb_E11-5_DelBs_r2_input.bw<br>ChIP_Rad21_DistalLimb_E11-5_FALLinv_r1.fq.gz ChIP_Rad21_DistalLimb_E11-5_FALLinv_r1.bw<br>ChIP_Rad21_DistalLimb_E11-5_FALLinv_r1_input.fq.gz ChIP_Rad21_DistalLimb_E11-5_FALLinv_r1_input.bw<br>ChIP_Rad21_DistalLimb_E11-5_FALLinv_r2.fq.gz ChIP_Rad21_DistalLimb_E11-5_FALLinv_r2.bw<br>ChIP_Rad21_DistalLimb_E11-5_FALLinv_r2_input.fq.gz ChIP_Rad21_DistalLimb_E11-5_FALLinv_r2_input.bw<br>ChIP_Rad21_DistalLimb_E11-5_FALL_r1.fq.gz ChIP_Rad21_DistalLimb_E11-5_FALL_r1.bw<br>ChIP_Rad21_DistalLimb_E11-5_FALL_r1_input.fq.gz ChIP_Rad21_DistalLimb_E11-5_FALL_r1_input.bw<br>ChIP_Rad21_DistalLimb_E11-5_FALL_r2.fq.gz ChIP_Rad21_DistalLimb_E11-5_FALL_r2.bw<br>ChIP_Rad21_DistalLimb_E11-5_FALL_r2_input.fq.gz ChIP_Rad21_DistalLimb_E11-5_FALL_r2_input.bw<br>ChIP_Rad21_DistalLimb_E11-5_R1F2_r1.fq.gz ChIP_Rad21_DistalLimb_E11-5_R1F2_r1.bw<br>ChIP_Rad21_DistalLimb_E11-5_R1F2_r1_input.fq.gz ChIP_Rad21_DistalLimb_E11-5_R1F2_r1_input.bw<br>ChIP_Rad21_DistalLimb_E11-5_R3-only_r1.fq.gz ChIP_Rad21_DistalLimb_E11-5_R3-only_r1.bw<br>ChIP_Rad21_DistalLimb_E11-5_R3-only_r1_input.fq.gz ChIP_Rad21_DistalLimb_E11-5_R3-only_r1_input.bw<br>ChIP_Rad21_DistalLimb_E11-5_RALL_r1.fq.gz ChIP_Rad21_DistalLimb_E11-5_RALL_r1.bw<br>ChIP_Rad21_DistalLimb_E11-5_RALL_r1_input.fq.gz ChIP_Rad21_DistalLimb_E11-5_RALL_r1_input.bw<br>ChIP_Rad21_DistalLimb_E11-5_RALL_r2.fq.gz ChIP_Rad21_DistalLimb_E11-5_RALL_r2.bw<br>ChIP_Rad21_DistalLimb_E11-5_RALL_r2_input.fq.gz ChIP_Rad21_DistalLimb_E11-5_RALL_r2_input.bw<br>ChIP_Rad21_DistalLimb_E11-5_WT_r1.fq.gz ChIP_Rad21_DistalLimb_E11-5_WT_r1.bw |
| **Genome browser session**<br>(e.g. UCSC) | https://genome.mdc-berlin.de/cgi-bin/hgTracks?<br>hgS_doOtherUser=submit&hgS_otherUserName=rdacemel&hgS_otherUserSessionName=DelBs_public |

## Methodology

| | |
|---|---|
| **Replicates** | One replicate was used for CTCF ChIP-seq experiments were used as a control to certify the total absence of CTCF binding upon  CTCF binding site deletions in the different mutant cell lines.  Rad21 ChIPmentation experiments were performed in duplicates as is the standard set by the ENCODE project for this type of experiments with the exception of R3-only and R1+F2 genotypes due to difficulties obtaining enough embryonic material from these genetic backgrounds. |
| **Sequencing depth** | ChIP_CTCF_mESC_DelBs // TOTAL: 53825880 // ALIGNED: 41999899 // LENGTH: 75bp // SINGLE-END<br>ChIP_CTCF_mESC_F1 // TOTAL: 53085169 // ALIGNED: 38545675 // LENGTH: 75bp // SINGLE-END<br>ChIP_CTCF_mESC_F2 // TOTAL: 51452800 // ALIGNED: 36878701 // LENGTH: 75bp // SINGLE-END<br>ChIP_CTCF_mESC_F-ALL // TOTAL: 44551297 // ALIGNED: 30035416 // LENGTH: 75bp // SINGLE-END<br>ChIP_CTCF_mESC_F-ALL-INV // TOTAL: 67485877 // ALIGNED: 46882387 // LENGTH: 75bp // SINGLE-END<br>ChIP_CTCF_mESC_R1 // TOTAL: 48777397 // ALIGNED: 33726017 // LENGTH: 75bp // SINGLE-END<br>ChIP_CTCF_mESC_R2 // TOTAL: 33328381 // ALIGNED: 19947596 // LENGTH: 75bp // SINGLE-END<br>ChIP_CTCF_mESC_R3 // TOTAL: 58086689 // ALIGNED: 43324716 // LENGTH: 75bp // SINGLE-END<br>ChIP_CTCF_mESC_R4 // TOTAL: 44628642 // ALIGNED: 30282667 // LENGTH: 75bp // SINGLE-END<br>ChIP_CTCF_mESC_R-ALL // TOTAL: 85948593 // ALIGNED: 66275753 // LENGTH: 75bp // SINGLE-END<br>ChIP_CTCF_mESC_ALL // TOTAL: 37638816 // ALIGNED: 20326201 // LENGTH: 75bp // SINGLE-END<br>ChIP_CTCF_DistalLimb_E11-5_WT_r1 // TOTAL:58028751 // ALIGNED:19189393 // LENGTH:75bp// SINGLE-END<br>ChIP_Rad21_DistalLimb_E11-5_ALL_r1 // TOTAL:58009791 // ALIGNED:45921400 // LENGTH:75bp// SINGLE-END<br>ChIP_Rad21_DistalLimb_E11-5_ALL_r1_input // TOTAL:62815605 // ALIGNED:40443783 // LENGTH:100bp// SINGLE-END |

ChIP_Rad21_DistalLimb_E11-5_ALL_r2 // TOTAL:40093948 // ALIGNED:30936732 // LENGTH:100bp // SINGLE-END
ChIP_Rad21_DistalLimb_E11-5_ALL_r2_input // TOTAL:41267431 // ALIGNED:41267431 // LENGTH:100bp // SINGLE-END
ChIP_Rad21_DistalLimb_E11-5_DelBs_r1 // TOTAL:50738061 // ALIGNED:37249790 // LENGTH:100bp // SINGLE-END
ChIP_Rad21_DistalLimb_E11-5_DelBs_r2 // TOTAL:55999774 // ALIGNED:41326409 // LENGTH:100bp // SINGLE-END
ChIP_Rad21_DistalLimb_E11-5_DelBs_r2_input // TOTAL:43255640 // ALIGNED:14296889 // LENGTH:100bp // SINGLE-END
ChIP_Rad21_DistalLimb_E11-5_FALLinv_r1 // TOTAL:56707612 // ALIGNED:45972724 // LENGTH:75bp // SINGLE-END
ChIP_Rad21_DistalLimb_E11-5_FALLinv_r1_input // TOTAL:42628140 // ALIGNED:29835034 // LENGTH:100bp // SINGLE-END
ChIP_Rad21_DistalLimb_E11-5_FALLinv_r2 // TOTAL:39116675 // ALIGNED:24580837 // LENGTH:100bp // SINGLE-END
ChIP_Rad21_DistalLimb_E11-5_FALLinv_r2_input // TOTAL:39519615 // ALIGNED:19168144 // LENGTH:100bp // SINGLE-END
ChIP_Rad21_DistalLimb_E11-5_FALL_r1 // TOTAL:59839080 // ALIGNED:45181541 // LENGTH:75bp // SINGLE-END
ChIP_Rad21_DistalLimb_E11-5_FALL_r1_input // TOTAL:65997517 // ALIGNED:45522134 // LENGTH:100bp // SINGLE-END
ChIP_Rad21_DistalLimb_E11-5_FALL_r2 // TOTAL:41343217 // ALIGNED:31073011 // LENGTH:100bp // SINGLE-END
ChIP_Rad21_DistalLimb_E11-5_FALL_r2_input // TOTAL:43250838 // ALIGNED:13358619 // LENGTH:100bp // SINGLE-END
ChIP_Rad21_DistalLimb_E11-5_R1F2_r1 // TOTAL:43417607 // ALIGNED:30083365 // LENGTH:100bp // SINGLE-END
ChIP_Rad21_DistalLimb_E11-5_R1F2_r1_input // TOTAL:38118414 // ALIGNED:18379675 // LENGTH:100bp // SINGLE-END
ChIP_Rad21_DistalLimb_E11-5_R3-only_r1 // TOTAL:51110374 // ALIGNED:37618702 // LENGTH:100bp // SINGLE-END
ChIP_Rad21_DistalLimb_E11-5_R3-only_r1_input // TOTAL:46269910 // ALIGNED:14783961 // LENGTH:100bp // SINGLE-END
ChIP_Rad21_DistalLimb_E11-5_RALL_r1 // TOTAL:50527290 // ALIGNED:38180577 // LENGTH:75bp // SINGLE-END
ChIP_Rad21_DistalLimb_E11-5_RALL_r1_input // TOTAL:64021077 // ALIGNED:40262844 // LENGTH:100bp // SINGLE-END
ChIP_Rad21_DistalLimb_E11-5_RALL_r2 // TOTAL:43654593 // ALIGNED:34243430 // LENGTH:100bp // SINGLE-END
ChIP_Rad21_DistalLimb_E11-5_RALL_r2_input // TOTAL:44550398 // ALIGNED:23027998 // LENGTH:100bp // SINGLE-END
ChIP_Rad21_DistalLimb_E11-5_WT_r1 // TOTAL:44702940 // ALIGNED:16362929 // LENGTH:100bp // SINGLE-END

Antibodies

CTCF Ab Diagenode:C15410210 Lot. No. A2359-00234D also stated in Material and Methods
RAD21 Ab (ABCAM) ab992 lot. num. GR3310168 also stated in Material and Methods

Peak calling parameters

No peak calling was performed since the objective of the ChIP-seq experiments was to certify the total absence of CTCF binding upon CTCF binding site deletions in the different mutant cell lines and the cohesin (Rad21) dynamics over a single locus, the EP-boundary.

Data quality

We manually assessed the absence of reads at the deleted CTCF binding sites, and the presence of the expected CTCF enrichment at the remaining and already described CTCF binding sites (see Extended Data Fig. 4). For Rad21 ChIPmentation we assessed the cohesin loading in the EP-boundary and its reproducibility between replicates (see UCSC session).

Software

bowtie v1.2.3
samtools v1.9
bedtools v2.29.2
bedGraphToBigWig (kentUtils) v4
deeptools (v3.5.1)

