## [Peer Review File · Nature Genetics]

Peer Review Information

Manuscript Title: In vivo dissection of a clustered-CTCF domain boundary reveals developmental principles of regulatory insulation

Corresponding author name(s): Dr Dario Lupianez

Reviewer Comments & Decisions:

Decision Letter, initial version:
--

9th Jun 2021

Dear Darío,

Your Article, "In vivo dissection of a clustered-CTCF domain boundary reveals developmental principles of regulatory insulation" has now been seen by 2 referees. Unfortunately, reviewer #3 did not submit a timely review, despite our multiple chase emails. We have now decided to proceed based on the two reports we received. I apologize for the long review process.

You will see from the reviewers' comments copied below that while they find your work of potential interest, some significant concerns are raised, which should be carefully addressed.

Reviewer #1 thinks that this is interesting and high-quality work. Their main request is that you perform additional ChIP-seq analysis of cohesin binding.

Reviewer #2 also says that this is well done and showcases a lot of potentially interesting data. Their main criticism is lack of mechanistic insight. This reviewer notes that loops formed by non-convergent CTCF-binding sites have been previously reported and that at this stage you don't provide a molecular explanation for your results.

In light of these comments, we cannot accept the manuscript for publication, but would be interested in considering a revised version that addresses these concerns.

If you choose to revise your manuscript taking into account all reviewer comments, please highlight all changes in the manuscript text file. At this stage we will need you to upload a copy of the manuscript in MS Word .docx or similar editable format.

We are committed to providing a fair and constructive peer-review process. Do not hesitate to contact me if there are specific requests from the reviewers that you believe are technically impossible or unlikely to yield a meaningful outcome.

*2) If you have not done so already please begin to revise your manuscript so that it conforms to our Article format instructions, available [here](http://www.nature.com/ng/authors/article_types/index.html). Refer also to any guidelines provided in this letter.

[REDACTED]

If you wish to submit a suitably revised manuscript we would hope to receive it within 6 months. If you cannot send it within this time, please let us know. We will be happy to consider your revision so long as nothing similar has been accepted for publication at Nature Genetics or published elsewhere. Should your manuscript be substantially delayed without notifying us in advance and your article is eventually published, the received date would be that of the revised, not the original, version.

Nature Genetics is committed to improving transparency in authorship. As part of our efforts in this direction, we are now requesting that all authors identified as 'corresponding author' on published papers create and link their Open Researcher and Contributor Identifier (ORCID) with their account on the Manuscript Tracking System (MTS), prior to acceptance. ORCID helps the scientific community achieve unambiguous attribution of all scholarly contributions. You can create and link your ORCID from the home page of the MTS by clicking on 'Modify my Springer Nature account'. For more information please visit please visit

<http://www.springernature.com/orcid>>www.springernature.com/orcid.

Thank you for the opportunity to review your work.

Sincerely,

Tiago

Tiago Faial, PhD
Senior Editor
Nature Genetics
<https://orcid.org/0000-0003-0864-1200>

Referee expertise:

Referee #1: 3D genomics

Referee #2: cohesin/CTCF function, 3D genomics

Referee #3: 3D genomics

Reviewers' Comments:

Reviewer #1:
Remarks to the Author:
Summary

In this study, Anania/Acemel et al. dissect the Epha4-Pax3 (EP) CTCF boundary, which contains 6 individual CTCF-binding sites (CBSs) and separates the Epha4 and Pax3 loci. The authors have previously shown that a large deletion in this region (DelB) including both the Epha4 gene and the EP boundary causes upregulation of Pax3, due to ectopic interactions with the Epha4 enhancers, resulting in limb malformation. However, a slightly smaller deletion (DelBs) which includes the Epha4 gene, but leaves the boundary intact, has no/little impact on gene expression and limb phenotype. The DelBs model therefore forms a very nice "minimal" system to further probe the function of the EP boundary, since the only thing that separates the Epha4 enhancers and the Pax3 gene is the EP boundary (i.e. there are no confounding effects from Epha4).

To dissect the function of the EP boundary, the authors generate 11 transgenic mouse models with homozygous deletions and inversions of the CBSs, in which they examine (1) gene expression with qPCR and WISH, (2) chromatin structure with capture-Hi-C and (3) limb phenotypes.

The EP boundary contains 2 forwardly oriented (F) and 4 reversely orientated (R) CBSs. The authors initially make individual deletions of all 6 CBSs. Only 3 of these result in a significant (but small) effect on gene expression ($\Delta R1$, $\Delta F1$ and $\Delta F2$). Interestingly, these are the CBSs with the strongest Cohesin occupancy. The authors then make combinatorial deletions of the R1+F2 sites and F1+F2 sites. In

both cases, the impact of these deletions has a much greater impact on insulation and gene expression than what would be expected based on adding up the effect of the individual deletions. This synergistic effect implies redundancy in function of individual CBSs. Moreover, the effect of these single and combinatorial deletions shows a quantitative correlation between reduction in boundary strength and impact on gene expression. Interestingly, the combined deletion of the forward CBSs also results in decreased loop strength of the centromeric loop, suggesting that F1 was responsible for forming this loop. This means that F1 forms this loop in a non-convergent manner. The authors propose that F1 is able to function as a non-convergent centromeric boundary, because it already binds a bulky Cohesin/CTCF complex as part of its convergent telomeric loop, which could form an obstacle for Cohesin extrusion from the centromeric side. This is an interesting hypothesis. The authors find further support for the existence of such simultaneous/paired non-convergent and convergent loops in available Hi-C data. To further confirm that F1 is responsible for maintaining the centromeric loop, the authors remove all reverse CBSs and show that the centromeric loop is maintained. This experiment also further confirms that the forward CBSs are responsible for most of the function of the EP boundary. To further test the impact of orientation and genomic context, the authors next ask whether they can “rescue” the ΔF -all model by flipping the E-P boundary without these two sites. In the ΔF -all-Inv model, the remaining R CBSs are in a forward orientation. Interestingly, this results in a very similar gene expression pattern and boundary strength as in the ΔF -all model. Together these experiments show that the function of a boundary is not only determined by the number of CBSs and their orientation, but that characteristics of individual sites can be more important. Finally, the authors remove all CBSs in the E-P boundary (ΔALL). This again results in a synergistic impact on gene expression compared to the ΔF -all and ΔR -all models. The impact on chromatin architecture and gene expression is slightly milder compared to the DelB model, which is likely due to the reduced distances between the Epha4 enhancers and the Pax3 gene in the DelB model.

Major Comments

In this manuscript, Anania/Acemel et al. address important questions about the mechanisms by which CBSs mediate insulation at TAD boundaries. In particular, they focus on the questions how multiple CBSs within a boundary cooperate to establish insulator function and what distinguishes a functional CBS from a non-functional CBS. To address these questions, they generated a very comprehensive set of boundary mutations, and present high-quality data to describe the impact of these mutations. Overall, I very much enjoyed reading this manuscript and I think it provides important new insights which will be of interest to many in the field.

The only major suggestion for improvement I have concerns the role of Cohesin. The authors show that there are only 3 CBSs which have an impact on gene expression when deleted individually. Each of these 3 CBSs are strongly co-bound by Cohesin. This suggests that Cohesin binding is an important determinant of boundary function, which is also in line with the loop extrusion model and other studies. In order to better interpret the molecular mechanisms of insulation in their models with boundary deletions and inversions, it would be very interesting to work out how these deletions change Cohesin occupancy in the EP boundary by performing ChIP-seq experiment for Cohesin. Potential shifts in Cohesin peaks in the EP boundary could strengthen the authors' conclusions about the role of redundancy, orientation, non-convergent paired loops, and overall give more insight into the molecular basis of insulator function. Moreover, since there is accumulating evidence that Cohesin loading is not random, perhaps the authors could also find out more about the locations where Cohesin is likely loaded onto chromatin based on these ChIP-seq data.

Minor Comments

1. I find it interesting that the interactions between the CTCF sites upstream of the Epha4 enhancers and downstream of the Pax3 gene appear to increase in some of the mutants with EP boundary deletions. Could the authors comment on this observation?
2. Some of the conclusions about redundancy are not entirely new. For example, both single and combined deletions of CBSs at the Shh and alpha-globin locus have been published previously (Williamson et al., Development, 2019; Paliou et al., PNAS, 2019; Hanssen et al., Nature Cell Biology, 2017). I think that this study provides a much more comprehensive analysis and brings many new insights, but I do think it would be good to relate these new findings to the earlier studies.
3. It is not clear if the Capture-Hi-C experiments have been performed in multiple replicates; if not, it would be nice if the authors could provide at least two independent replicates for their key mutants.
4. Why are the CTCF tracks in Figure 1B and Supp. Figure 3A different? They seem to be from the same tissue, background, and reference.
5. Fig 5C – it would be nice to highlight the DelB deletion, since this might not be obvious to everyone.
6. If the authors would wish to further investigate the impact of CBS orientation, I think it would be very interesting to also assess an ΔR -all-inv model. However, I want to stress that I think that the authors already did a very thorough job and made a very comprehensive set of boundary deletions and inversions, so I do not think this should be required for publication.

Reviewer #2:

Remarks to the Author:

Anania et al. present manuscript in which they systematically analyze a boundary region between the Pax3 and Epha4 gene, formed by 6 CTCF sites in different orientations (2 forward and 4 reverse). They show that mutation of individual CTCF binding sites is (supposedly) buffered by the presence of nearby CTCF sites in the same orientation. Even when all CTCF sites in a specific orientation are mutated, there is still enhancer blocking activity. When only 2 forward sites are left (ΔR -ALL) the insulating capacity of the border is stronger compared to 4 reverse (ΔF -ALL), both on the 3D genome and gene expression. Interestingly, when the entire border is inverted, the effect on insulation is limited. Deleting all CTCF sites, show a synergistic effect on the misexpression of Pax3.

The manuscript represents a tremendous amount of genetic data. The experiment data is of high quality and is presented well. Although the described phenomena are interesting and shed some new light on how a boundary could function, it does not explain how it actually could work. This is unfortunate, because now it mostly feels like a summary of interesting mutants without actually providing insight into how individual CTCF sites contribute to enhancer blocking. Likely, the two reverse sites are strong boundary elements (compared to the forward sites), but we are left wondering why. Mechanistic insight into why certain CTCF sites form strong boundaries and others do not, would strengthen the manuscript.

Other points:

- * The data is extremely complex and difficult to understand: it would already help a lot if the zoom-ins would also contain the position and orientation of the CTCF sites that for the non-EP anchors of the loops that are shown.
- * The idea that loops can be formed by non-convergent CTCF-sites has already been shown using multi-contact 4C in Allahyar et al. 2018. The idea of loop interference is conceptually the same the loop collisions described in that paper.
- * In Supp. Fig. 9D, please show that cumulative distribution function of the log₁₀-transformed loop lengths for the different categories, so that there can be better comparison.
- * The conclusion that the boundary score is decreased by 19% is incorrect. Because this is a log-score one cannot simple calculate percentages.

Reviewer #3:

None

Decision on Nature Genetics submission NG-A57461
Message: 9th Jun 2021

Dear Darío,

Reviewer #3 sent me some comments today (please see below) after I informed them of our decision. I think the feedback is thoughtful and I ask that you incorporate it in the revised manuscript. Thanks.

Best wishes,

Tiago

Reviewer #3:

The quantitative modulations of digit length as a function of deletions of single CTCF sites shown in Figure 6 is a fantastic example of how noncoding mutations can have a quantitative impact on physiological (and not necessarily pathological) states.

I wonder why the DR1 mutant was missing in the correlation in panel 6D. I would ask to check by ChIP if the CTCF sites that do *not have cohesin stalled in the DelB background start to stop cohesin when the other 'primary' sites are deleted. This would add an entirely new and relevant layer of information on what makes CTCF sites important for insulation (also adding to the sequence-specific insulation difference detected in Bing Ren's recent Nature Genetics paper, I believe).

Another point is that there is sometimes a slight confusion between 'functional insulation' and 'structural insulation' and in some instances these concepts are used interchangeably. For example, in the conclusions, the sentence "Overall, these results illustrate how boundary insulation strength can serve as a modulator of gene expression and developmental phenotypes, by allowing permissive functional interactions between neighboring TADs" is a bit misleading because the authors did not provide insulation score changes for single CTCF site mutants; I think that 'insulation strength' here means 'number of CTCF sites'.

The other minor thing is that there is no indication in the main text or supplementary information of how the differential Hi-C heatmaps were calculated – I had to go back to the authors' previous paper (Bianco et al.) to find out. It's however an important piece of information to relate the transcriptional changes the authors see to changes in (normalized) contact probabilities.

Author Rebuttal to Initial comments

Anania, Acemel et al. Response to reviewers:

Reviewer #1:

In this manuscript, Anania/Acemel et al. address important questions about the mechanisms by which CBSs mediate insulation at TAD boundaries. In particular, they focus on the questions how multiple CBSs within a boundary cooperate to establish insulator function and what distinguishes a functional CBS from a non-functional CBS. To address these questions, they generated a very comprehensive set of boundary mutations, and present high-quality data to describe the impact of these mutations. Overall, I very much enjoyed reading this manuscript and I think it provides important new insights which will be of interest to many in the field.

We thank the reviewer for the positive appreciation of our work

The only major suggestion for improvement I have concerns the role of Cohesin. The authors show that there are only 3 CBSs which have an impact on gene expression when deleted individually. Each of these 3 CBSs are strongly co-bound by Cohesin. This suggests that Cohesin binding is an important determinant of boundary function, which is also in line with the loop extrusion model and other studies. In order to better interpret the molecular mechanisms of insulation in their models with boundary deletions and inversions, it would be very interesting to work out how these deletions change Cohesin occupancy in the EP boundary by performing ChIP-seq experiment for Cohesin. Potential shifts in Cohesin peaks in the EP boundary could strengthen the authors' conclusions about the role of redundancy, orientation, non-convergent paired

loops, and overall give more insight into the molecular basis of insulator function. Moreover, since there is accumulating evidence that Cohesin loading is not random, perhaps the authors could also find out more about the locations where Cohesin is likely loaded onto chromatin based on these ChIP-seq data.

We agree that cohesin dynamics are fundamental to better understand how the EP-boundary functions. Accordingly, we have performed ChIPmentation using the RAD21 antibody in embryonic e11.5 distal limbs in all of our mutant backgrounds and wild type mice.

In contrast to what is observed on ChIP-seq experiments from Andrey et al, 2017 (previous Extended Data Fig. 3), our new ChIPmentation experiments show that 2 of the other EP boundary CBS (R2 and R3) are able to stall cohesin (new Extended Data Fig. 3). We believe that this effect is mainly derived from technical differences in the protocol (i.e. our ChIPmentation protocol uses less sonication cycles, which may preserve low affinity binding better). Nevertheless, cohesin accumulation still remains higher in the CBS that were proven to be more critical for EP boundary stability (namely F1, F2 and R1). Extended Data Figure 3 and the main text have been modified accordingly to show these results.

As depicted in Extended Data.Fig. 9, while cohesin accumulation is abrogated in deleted CBS, it is not obviously redistributed in the remaining CBS of the boundary. Thus, the absence of major compensatory effects is in agreement with the progressive decrease of EP-boundary insulation observed in combined CBS deletions. We only observed some cohesin occupancy changes in the F1 site in the $\Delta R1F2$ mutant, like due to the deletion of the other forward-oriented CBS (F2). Overall, we find that cohesin occupancy on convergently-oriented CBS is highly predictive of the presence of loops. But we find one notable exception in the $\Delta R-ALL$ mutant, in which the cohesin at F1 and F2 CBS also mediate non-convergent loops with the cohesin-occupied CBS at the centromeric boundary of the *Epha4* TAD.

All these new experiments are detailed in Extended Data Fig. 3 and Extended Data. Fig. 9, as well as described in the main text.

Minor Comments

1. I find it interesting that the interactions between the CTCF sites upstream of the Epha4 enhancers and downstream of the Pax3 gene appear to increase in some of the mutants with EP boundary deletions. Could the authors comment on this observation?

We thank the reviewer for raising this point. In this case, this newly-formed loop connects the *Epha4* TAD centromeric and the *Pax3* telomeric boundaries when the EP-boundary is compromised. We refer to this loop as a “*meta-TAD loop*” as it is probably linked to the meta-TAD structures described in Fraser et al, 2015. Our interpretation is that the cohesin complex is stalled at lower rates at the perturbed EP-boundary (as observed in the new RAD21 ChIP data from Extended Data Fig 9), allowing the complex to progress until the next CBS. Therefore, the degree of interactions between the outer *Epha4* and *Pax3* boundaries increases and the corresponding meta-TAD loop becomes more prominent. We have now described this observation in the main text. We have also added an extra row to Extended Data. Fig. 8 with zoom-ins of the meta-TAD loop region in all different mutants. This figure shows that the strength of the meta-TAD loop increases when the boundary is compromised.

2. Some of the conclusions about redundancy are not entirely new. For example, both single and combined deletions of CBSs at the *Shh* and alpha-globin locus have been published previously (Williamson et al., *Development*, 2019; Paliou et al., *PNAS*, 2019; Hanssen et al., *Nature Cell Biology*, 2017). I think that this study provides a much more comprehensive analysis and brings many new insights, but I do think it would be good to relate these new findings to the earlier studies.

We apologize for not including these studies in our main discussion. Following the reviewer's suggestion, we have expanded the discussion accordingly, to explicitly relate our findings with those reported at the *Shh* and the *alpha-globin* locus.

As the field is progressing rapidly, we have also included additional references that are of relevance for our study. In Amândio et al., *Genes & Development*, 2021, the authors report that the *HoxD* centromeric and telomeric TADs do not fuse upon deletion of all forward-oriented CBS. Chakraborty et al., *BioRxiv*, 2021 provides interesting insights on the interplay between enhancer strength and insulation from a quantitative perspective.

3. It is not clear if the Capture-Hi-C experiments have been performed in multiple replicates; if not, it would be nice if the authors could provide at least two independent replicates for their key mutants.

cHi-C experiments were performed in single-replicates using a pool of distal limbs from different embryos. These are commonly accepted sample sizes for these types of experiments (Franke et al., *Nature*, 2016; Bianco et al., *Nature Genetics* 2018; Rouco et al., *Nature Communications*, 2021;). cHi-C experiments have been shown to be highly reproducible, so we leveraged on the fact that changes between different mutant

backgrounds are expected to be local and circumscribed to the *Epha4* and *Pax3* TADs (1.5 Mb aprox). Since the total captured region comprises 10Mb, we have calculated correlations between the different experiments outside of the *Epha4* and *Pax3* TADs, to assess for technical variability. Pairwise Pearson's correlations were high (>0.89 in all cases; **Reviewer's Fig. 1**) highlighting the reproducibility of the presented cHi-C experiments. This is described in the Methods section, but we updated it to include the correlations assesment including the new $\Delta R3$ -only mutant.

Reviewer's Fig. 1. Assesment of cHi-C reproducibility

The matrix displays Pearson's correlations between cHi-C experiment from different mutants, corresponding to the regions outside of the *Epha4* and *Pax3* TADs within the 8.5 Mb included in the cHi-C experiments.

4. Why are the CTCF tracks in Figure 1B and Supp. Figure 3A different? They seem to be from the same tissue, background, and reference.

We apologize to the reviewer for the misunderstanding. We employed different datasets: Rodriguez-Carballo *et al.*, *Genes & Development*, 2017 for Fig. 1B, and Andrey *et al.*, *Genome Research*, 2017. Nevertheless, we have substituted these panels from our own datasets in both Fig. 1B and Extended Data Fig. 3A (CTCF and RAD21 ChIP in wild type distal limb buds).

5. Fig 5C – it would be nice to highlight the DelB deletion, since this might not be obvious to everyone.

The *DelB* deletion location is now indicated in Fig. 6E.

*6. If the authors would wish to further investigate the impact of CBS orientation, I think it would be very interesting to also assess an ΔR -all-*inv* model. However, I want to stress that I think that the authors already did a very thorough job and made a very comprehensive set of boundary deletions and inversions, so I do not think this should be required for publication.*

We thank the reviewer for the suggestion. We performed the inversion experiments to determine the impact of the surrounding genomic contexts on the function of the EP boundary. We revealed that the surrounding context has a minimal influence on boundary function and, based on these results, we might expect a similar outcome on an ΔR -all-*inv* model. Nevertheless, it could be interesting to determine if F1 and F2 are still able to form non-convergent loops in an inverted orientation. While we have not performed these experiments for the current study, it is a question that we might pursue in the near future.

Reviewer #2:

The manuscript represents a tremendous amount of genetic data. The experiment data is of high quality and is presented well. Although the described phenomena are interesting and shed some new light on how a boundary could function, it does not explain how it actually could work. This is unfortunate, because now it mostly feels like a summary of interesting mutants without actually providing insight into how individual CTCF sites contribute to enhancer blocking. Likely, the two reverse sites are strong boundary elements (compared to the forward sites), but we are left wondering why. Mechanistic insight into why certain CTCF sites form strong boundaries and others do not, would strengthen the manuscript.

We completely share the reviewer's opinion about the importance of elucidating the mechanisms that determine the functionality of individual CBS. However, we feel that this important question falls beyond the scope of this paper, which is focused on exploring the functional interplay between CBS and their effects on transcription and phenotypes. Elucidating those mechanisms likely requires a different setup than the one taken by us. For example, a recent paper by the Ren's lab (Huang *et al.*, *Nature Genetics*, 2021) employed an *in vitro* insulator reporter assay to tackle this question, finding that the genomic regions that surround CBS might contribute to its functionality

Nevertheless, we performed additional genome-wide analyses to determine whether the unique properties of bidirectional-looping CBS could be explained by differences in cohesin binding, open chromatin, conservation or MAZ binding (Xiao *et al.* *PNAS*, 2021; Ortabozkoyun *et al.* *Nature*, 2022). However, all these analyses did not yield any obvious relationship (**Reviewer's Fig. 2**)

Reviewer's Fig. 2. Bidirectional loop features. Coverage plots centered CTCF motifs inside CBS from mouse embryonic stem cells (CTCF binding from Bonev et al. 2017, GSE96107). **A.** PhastCons scores derived from 60 vertebrate species. Multiz alignment were downloaded from UCSC. **B.** ATAC-seq in mouse stem cells are from Bulut-Karslioglu et al. 2018 (GSE98358). **C.** Rad21 ChIP-seqs in mouse stem cells are from Kaaij et al. 2019 (GSE125129). **D.** MAZ binding in mouse stem cells are from Ortabozkoyun et al. 2021 (GSE190991).

In addition, to gain further insights on the interplay between CBS, we have generated a new mutant ($\Delta R3$ -only). The experiments performed in this mutant background revealed two novel mechanistic insights. First, the functionality of R3, which displays the highest CTCF occupancy within the CBS cluster, is not increased in the absence of other sites. This again points to the existence of other factors, besides CTCF, required for CBS robust insulation.

And second, that CBS with decreased insulatory function might be insufficient to block enhancer-promoter communication, even if they form loops. We find this an interesting observation, as the configuration observed in ChIP-seq from the $\Delta R3$ -only mutant (a large TAD with an internal loop) resembles the 3D structure of several TADs across the genome. In those cases, internal loops might not be sufficient to block the functional interactions between enhancers and genes located at both sides of the loop.

We have included these new results in Extended Data Fig 12 and modified the main text accordingly. We hope that these expanded findings on how CBS interact to build robust TAD boundaries will serve to reinforce the scope of the study.

Other points:

** The data is extremely complex and difficult to understand: it would already help a lot if the zoom-ins would also contain the position and orientation of the CTCF sites that for the non-EP anchors of the loops that are shown.*

We agree with the reviewer that adding the missing CBS position and orientation will be helpful for the reader. We have now added this information in the corresponding figures. We have also added a new schematic to facilitate the interpretation of the zoom-ins (Extended Data. Fig. 8A).

** The idea that loops can be formed by non-convergent CTCF-sites has already been shown using multi-contact 4C in Allahyar et al. 2018. The idea of loop interference is conceptually the same the loop collisions described in that paper.*

We thank the reviewer for this comment and apologize for not including this study in the discussion.

Overall, our findings are complementary to the results reported in Allahyar *et al.*, *Nature Genetics*, 2018. In this study, cohesin stalling is determined based on the analyses performed in *WAPL* mutants, in which cohesin dynamics are perturbed. Our results extend these results to a wild-type context, as well as providing a functional validation for this phenomenon (ΔF -all and ΔR -all).

We have now updated the discussion so that our results are better contextualized in the light of those previous findings.

** In Supp. Fig. 9D, please show that cumulative distribution function of the log10-transformed loop lengths for the different categories, so that there can be better comparison.*

We thank the reviewer for the visualization suggestion, we have accordingly updated Extended Data Fig. 9D (now Extended Data. Fig 10) as suggested.

** The conclusion that the boundary score is decreased by 19% is incorrect. Because this is a log-score one cannot simple calculate percentages.*

We apologize for this erroneous interpretation. We have now explicitly stated boundary score values throughout the text, instead of such percentages.

Reviewer #3:

The quantitative modulations of digit length as a function of deletions of single CTCF sites shown in Figure 6 is a fantastic example of how noncoding mutations can have a quantitative impact on physiological (and not necessarily pathological) states.

We are thankful to the reviewer for its positive appreciation of our study.

I wonder why the DR1 mutant was missing in the correlation in panel 6D.

We did not perform Capture Hi-C in this particular background and, therefore, insulation score could not be calculated. The moderate *Pax3* misexpression on $\Delta R1$ mutants suggests that the structural changes should be subtle, with a considerable structural preservation of the two TADs.

*I would ask to check by ChIP if the CTCF sites that do *not have cohesin stalled in the DelB background start to stop cohesin when the other 'primary' sites are deleted. This would add an entirely new and relevant layer of information on what makes CTCF sites important for insulation (also adding to the sequence-specific insulation difference detected in Bing Ren's recent Nature Genetics paper, I believe).*

We thank the reviewer for this valuable comment. As also requested by Reviewer #1, we have performed RAD21 ChIP seq in all mouse mutants. As described in the reply to reviewer 1, we do not observe compensatory accumulation of RAD21 in adjacent CBS was identified in the different mutants, with the exception of F1 in the $\Delta R1F2$ background. The overall absence of compensatory effects is in agreement with the progressive decrease of EP-boundary insulation observed in combined CBS deletions.

Another point is that there is sometimes a slight confusion between 'functional insulation' and 'structural insulation' and in some instances these concepts are used interchangeably. For example, in the conclusions, the sentence "Overall, these results illustrate how boundary insulation strength can serve as a modulator of gene expression and developmental phenotypes, by allowing permissive functional interactions between neighboring TADs" is a bit misleading because the authors did not provide insulation score changes for single CTCF site mutants; I think that 'insulation strength' here means 'number of CTCF sites'.

This is an important punctualization. However, we want to clarify that we intended to refer here to *Pax3* ectopic expression in the mutant series $\Delta R1F2$, ΔR -all, ΔF -all, Δall , *DelB* and the new $\Delta R3$ -only mutant for which we have cHi-C data and indeed insulation scores calculated.

We have examined the main text to ensure that such confusion is avoided.

The other minor thing is that there is no indication in the main text or supplementary information of how the differential Hi-C heatmaps were calculated – I had to go back to the authors' previous paper (Bianco et al.) to find out. It's however an important piece of information to relate the transcriptional changes the authors see to changes in (normalized) contact probabilities.

Differential cHi-C map calculation is described in the methods sections: *Briefly, first the coverage of the matrices to be subtracted was equalized dividing by the total number of reads. Then, the two matrices were subtracted element-wise and each value of the subtraction was converted to a z-score taking into account the rest of values belonging to the same sub-diagonal (corresponding to interactions happening at equivalent genomic distances).*

The custom python code used to implement these subtraction method is also available in the gitlab repository: <https://gitlab.com/rdacemel/anania2021/-/blob/master/notebooks/3.1-CaptureC.ipynb>

Decision Letter, first revision:

Our ref: NG-A57461R

29th Mar 2022

Dear Darío,

Thank you for submitting your revised manuscript entitled "In vivo dissection of a clustered-CTCF domain boundary reveals developmental principles of regulatory insulation" (NG-A57461R). It has now been seen by the original referees and their comments are below. The reviewers find that the paper has improved in revision, and therefore we'll be happy in principle to publish it in Nature Genetics, pending minor revisions to satisfy the referees' final requests and to comply with our editorial and formatting guidelines.

We will be performing detailed checks on your paper and will send you a checklist detailing our editorial and formatting requirements soon. Please do not upload the final materials and make any revisions until you receive this additional information from us.

Thank you again for your interest in Nature Genetics. Please do not hesitate to contact me if you have any questions.

Congratulations!

Sincerely,

Tiago

Tiago Faial, PhD
Senior Editor
Nature Genetics
<https://orcid.org/0000-0003-0864-1200>

Reviewer #1 (Remarks to the Author):

I appreciate the extra experiments the authors have performed for their revision, which have addressed my concerns. I recommend this manuscript for publication in Nature Genetics, as I think it provides important new insights which will be of interest to the field.

Reviewer #2 (Remarks to the Author):

The authors have addressed all my comments.

One of the important conclusions of the study by Allahyar et al. is that loop collisions are not only prevalent in WAPL KO cells, but also in wild-type cells. I have reproduced the relevant section below:

“We [...] conclude that loop collision and anchor aggregation also occur in WT cells, but less frequently, as a result of the counteracting effect of WAPL (Fig. 4e,f and Supplementary Fig. 13).”

I urge the authors to change the text to be consistent with previously published literature.

Reviewer #3 (Remarks to the Author):

The authors have addressed most of my previous comments. This already elegant manuscript is further strengthened by the addition of cohesin ChIP-seq data. While it would be important to address the reasons behind the absence of compensation by remaining CTCF sites in future work, I agree with the authors that this goes beyond the scope of their study.

There seems to be however some residual confusion between functional and structural insulation in some passages in the manuscript. This is particularly evident in the text related to the new Suppl. Figure 12: "...demonstrating that CBS with robust insulator function can create chromatin loops independently of their CTCF motif orientation". It is unclear which insulator function the authors attribute to this CTCF site: Do they mean that this is a CTCF site that has strong 'transcriptional insulation' function or rather 'structural insulation function' - or just that it leads to an accumulation of cohesin in the absence of other sites? It would be important to avoid any possible confusion.

Author Rebuttal, first revision:

Anania, Acemel et al. Response to reviewers:

Reviewer #1 (Remarks to the Author):

I appreciate the extra experiments the authors have performed for their revision, which have addressed my concerns. I recommend this manuscript for publication in Nature Genetics, as I think it provides important new insights which will be of interest to the field.

We thank Reviewer #1 for their comments and insight and we are pleased that they recommend the manuscript for publication.

Reviewer #2 (Remarks to the Author):

The authors have addressed all my comments.

One of the important conclusions of the study by Allahyar et al. is that loop collisions are not only prevalent in WAPL KO cells, but also in wild-type cells. I have reproduced the relevant section below:

“We [...] conclude that loop collision and anchor aggregation also occur in WT cells, but less frequently, as a result of the counteracting effect of WAPL (Fig. 4e,f and Supplementary Fig. 13).”

I urge the authors to change the text to be consistent with previously published literature.

We apologize to Reviewer #2 for our inaccurate reproduction of Allahyar et al. conclusions. We have corrected the sentence accordingly: *“A similar phenomenon, termed loop collision, has been observed in cells depleted from the cohesin-releaser factor WAPL and, to a lesser degree, in wildtype cells. Our results extend those findings, constituting an in vivo experimental validation for a scenario predicted by the loop extrusion model.”*

We also want to thank the reviewer for the comments and insights during the peer review process.

Reviewer #3 (Remarks to the Author):

The authors have addressed most of my previous comments. This already elegant manuscript is further strengthened by the addition of cohesin ChIP-seq data. While it would be important to address the reasons behind the absence of compensation by remaining CTCF sites in future work, I agree with the authors that this goes beyond the scope of their study.

There seems to be however some residual confusion between functional and structural insulation in some passages in the manuscript. This is particularly evident in the text related to the new Suppl. Figure 12: "...demonstrating that CBS with robust insulator function can create chromatin loops independently of their CTCF motif orientation". It is unclear which insulator function the authors attribute to this CTCF site: Do they mean that this is a CTCF site that has strong 'transcriptional insulation' function or rather 'structural insulation function' - or just that it leads to an accumulation of cohesin in the absence of other sites? It would be important to avoid any possible confusion.

We agree with Reviewer #3 that the phrasing of the aforementioned sentence could be misinterpreted. We have simplified the sentence to avoid this "...demonstrating that specific CBS can create chromatin loops independently of their motif orientation, through loop interference". We have also revised the whole manuscript in order to clarify potential confusions related to the term insulating function and finding the following sentences:

- *"Accordingly, the boundary score of the EP boundary in $\Delta R1+F2$ mutants was decreased to $BS=0.82$ compared to $BS=0.99$ in DelBs, reflecting a weakened insulating capacity"* where we have now clarified that we refer to structural insulation: *"Accordingly, the boundary score of the EP boundary in $\Delta R1+F2$ mutants was decreased, reflecting a weakened **structural insulation**"*
- *"... as enhancer-promoter cross-talk and gene activation are largely proportional to boundary insulation strength"* to *"as enhancer-promoter cross-talk and gene activation correlate with the strength of structural insulation conferred by boundaries"*

In other occasions, the term insulation is used without clarifying whether it is structural, functional or both. However, in all of those cases we refer to both structural and functional insulation describing mutants for which we have both cHi-C and Pax3 expression data. We hope that the changes made improve the accuracy and readability of the manuscript. Therefore, we want to thank the Reviewer #3 for this punctualization and for all of the past feedback throughout the peer review process.

Final Decision Letter

In reply please quote: NG-A57461R1 Lupiáñez

31st May 2022

Dear Darío,

I am delighted to say that your manuscript "In vivo dissection of a clustered-CTCF domain boundary reveals developmental principles of regulatory insulation" has been accepted for publication in an upcoming issue of Nature Genetics.

Your paper will be published online after we receive your corrections and will appear in print in the next available issue. You can find out your date of online publication by contacting the Nature Press Office (press@nature.com) after sending your e-proof corrections. Now is the time to inform your Public Relations or Press Office about your paper, as they might be interested in promoting its publication. This will allow them time to prepare an accurate and satisfactory press release. Include your manuscript tracking number (NG-A57461R1) and the name of the journal, which they will need when they contact our Press Office.

Acceptance is conditional on the data in the manuscript not being published elsewhere, or announced in the print or electronic media, until the embargo/publication date. These restrictions are not

intended to deter you from presenting your data at academic meetings and conferences, but any enquiries from the media about papers not yet scheduled for publication should be referred to us.

Please note that *Nature Genetics* is a Transformative Journal (TJ). Authors may publish their research with us through the traditional subscription access route or make their paper immediately open access through payment of an article-processing charge (APC). Authors will not be required to make a final decision about access to their article until it has been accepted. [Find out more about Transformative Journals](https://www.springernature.com/gp/open-research/transformative-journals)

Authors may need to take specific actions to achieve [compliance](https://www.springernature.com/gp/open-research/funding/policy-compliance-faqs) with funder and institutional open access mandates. If your research is supported by a funder that requires immediate open access (e.g. according to [Plan S principles](https://www.springernature.com/gp/open-research/plan-s-compliance)) then you should select the gold OA route, and we will direct you to the compliant route where possible. For authors selecting the subscription publication route, the journal's standard licensing terms will need to be accepted, including [self-archiving and license to publish](https://www.nature.com/nature-portfolio/editorial-policies/self-archiving-and-license-to-publish). Those licensing terms will supersede any other terms that the author or any third party may assert apply to any version of the manuscript.

Please note that Nature Portfolio offers an immediate open access option only for papers that were first submitted after 1 January, 2021.

Sincerely,

Tiago

Tiago Faial, PhD
Senior Editor
Nature Genetics
<https://orcid.org/0000-0003-0864-1200>